# SENSIBLE ADVERSARIAL LEARNING

## ABSTRACT

The trade-off between robustness and standard accuracy has been consistently reported in the machine learning literature. Although the problem has been widely studied to understand and explain this trade-off, the problem seems to remain as an open problem. In this paper, motivated by the fact that the high dimensional distribution is poorly represented by limited data samples, we introduce sensible adversarial learning and demonstrate the synergistic effect between pursuits of natural accuracy and robustness. Specifically, we define a sensible adversary which is useful for learning a defense model and keeping a high natural accuracy simultaneously. We theoretically establish that the Bayes rule is the most robust multi-class classifier with the 0-1 loss under sensible adversarial learning. We propose a novel and efficient algorithm that trains a robust model with sensible adversarial examples, without a significant drop in natural accuracy. Our model on CIFAR10 yields state-of-the-art results against various attacks with perturbations restricted to $\ell_\infty$ with $\epsilon = 8/255$, e.g., the robust accuracy 57.23% against PGD attacks as well as the natural accuracy 91.51%.

## 1 INTRODUCTION

With many impressive successes of deep learning, there are a multitude of applications of deep neural networks (DNNs) that permeate in our everyday life. As DNNs are applied in security-critical systems such as malware detection, face identification, and autonomous driving, robustness of DNNs against adversarial attacks, i.e., the intently perturbed inputs to fool the system, has become an important research topic (Szegedy et al., 2013; Papernot et al., 2016; Biggio et al., 2013).

One of the most widely studied classes of adversarial perturbations is $\ell_p$-norm constrained adversarial perturbations (Szegedy et al., 2013). Madry et al. (2017) formalize the adversarial learning against this class of perturbations as a minimization problem of adversarial risk defined in a following way. Let $(X, y) \in \mathcal{X} \times \mathcal{Y}$ be from some unknown distribution $\mathbb{P}_{X,Y}$. Given a loss function $\ell : \mathcal{Y} \times \mathcal{Y} \to \mathbb{R}$ and a constraint constant $\epsilon > 0$, the adversarial robust risk is

$$\mathcal{R}_{rob}(f) = \mathbb{E}_{X,Y} \big[ \max_{\|\delta\|_p \leq \epsilon} \ell(f(X + \delta), y) \big]. \tag{1}$$

Many adversarial learning methods can be interpreted as empirical minimization of (1) (Goodfellow et al., 2014; Kurakin et al., 2016b; Ruitong Huang & Szepesvari, 2015; Madry et al., 2017). For this optimization problem, Madry et al. (2017) propose to train a robust model with the augmented data generated by the projected gradient descent method (PGD). On this adversarial training, they make two important observations. First, it costs natural accuracy. A network trained with adversarial examples tends to have a lower natural accuracy than a naturally trained network. This trade-off is observed even with a small training $\epsilon$. Second, the adversarial training requires a larger model capacity than the natural training does. If the model capacity is only sufficient for the natural learning, the adversarial training can converge to a constant function.

For a large $\epsilon$, the optimization problem of (1) itself may pose the trade-off. For instance, Tsipras et al. (2018) show an example of an inherent tension between pursuits of accuracy and robustness when $\epsilon$ is large enough to change the true class. For a smaller $\epsilon$, however, the formulation in (1) does not explicitly pose any conflict between the pursuit of robustness and accuracy. Note that $\mathcal{R}_{rob}(f)$ is an upper bound of the standard risk of $f$. A perfectly robust model $f$ with $\mathcal{R}_{rob}(f) = 0$ is also perfectly accurate for natural learning. If the perfect classifier exists in a given model class, the trade-off may be caused by the large sample complexity of adversarially robust generalization

(Schmidt et al., 2018; Yin et al., 2018; Stutz et al., 2019). Without sufficiently large amount of data, the empirical minimization of (1) may result in a large standard risk by converging to a model of a poor robust risk. On the other hand, if robust learning converges to a constant function, it cannot achieve natural accuracy. In this sense, to resolve the trade-off problem, we may need to deal with the increased requirement on the model capacity.

In this paper, we propose a novel framework, called ***sensible adversary***, in order to overcome the trade-off between natural accuracy and robustness. In particular, we restrict adversarial perturbations not to cross the Bayes decision boundary besides the $\epsilon$-ball constraint, so that the perturbation ball is adaptively modified for every single data point. Our main contributions are:

- Under the framework of ***sensible adversary***, the pursuit of robustness and accuracy given an enough model capacity can align with each other, i.e., there is no trade-off. We theoretically establish the Bayes rule is most robust against the sensible adversary. If the Bayes decision boundary can be far from data manifolds at least by $\epsilon$, our pursuit of sensible robustness does not cost any adversarial robust risk.

- We propose an efficient algorithm for *sensible adversarial training* , which utilizes sensible adversaries in the absence of the true Bayes rule. This sensible adversarial training enjoys robustness without a significant drop of natural accuracy. Furthermore, the algorithm is not sensitive to the model capacity. When insufficient model capacity is given, our algorithm does not collapse to a constant function. Instead, it trains a model as robust as possible.

- We experimentally demonstrate that sensible adversarial training enables to stably learn a robust and accurate model. In particular, on CIFAR10, we achieve $91.51\%$ natural test accuracy and $57.23\%$ robust test accuracy against $\ell_\infty$ PGD attacks constrained to $\epsilon = 8/255$. To the best of our knowledge, there is no approaches known to achieve natural accuracy more than $90\%$, while achieving more than $55\%$ of robust accuracy against PGD attacks of $\epsilon = 8/255$. Moreover, no previous approaches pursuing robustness against this attack achieved the natural accuracy more than .

## 1.1 RELATED WORK

Madry et al. (2017) formalize adversarial learning as a mini-max problem given perturbation restriction, and theoretically and empirically established the feasibility of the optimization. Our sensible adversary redefines the set of perturbation in the inner maximization problem on which their theoretical result is directly applicable. Tsipras et al. (2018) investigate the possible source of robust trade-off. The key idea is that when there are features that are useful for natural classification but vulnerable to adversarial perturbations, a robust model would abandon these features because otherwise all of these features can adversarially move to promote incorrect prediction. Our work explores the possibility of learning a robust model while not allowing such collective adversarial migration. While a class change by adversarial examples typically has been prevented by using a *small $\epsilon$*, Suggala et al. (2018) explicitly ignore an adversarial perturbation that crosses the decision boundary of a Bayes rule. Zhang et al. (2019) also investigate the Bayes decision boundary to resolve the trade-off problem. They search for a model $f$ having a small weighted sum of the natural risk and a probability that an adversarial example can cross the decision boundary of $f$. Gilmer et al. (2018) show that in high dimensional setting, even small test error can imply the existence of adversarial examples for most of data points. Our effort to prioritize natural accuracy to find a robust model is consistent to the view in Gilmer et al. (2018). More related work will be presented in Appendix A.

## 1.2 NOTATION

Let $\mathcal{F}$ denote the class of functions represented by DNNs with a fixed architecture. An optimal robust model w.r.t. $\mathbb{P}_{X,Y}$ is denoted by $f_{rob} = \arg\min_{f \in \mathcal{F}} \mathcal{R}_{rob}(f)$. Denote the standard risk w.r.t $\mathbb{P}_{X,Y}$ by $\mathcal{R}_{std}(f) = \mathbb{E}_{X,Y}[\ell(f(X),y)]$ and its optimal natural model by $f_{std} = \arg\min_{f \in \mathcal{F}} \mathcal{R}_{std}(f)$. Let $\tilde{\mathbb{P}}_{X,Y} = \mathbb{P}_{X,Y}|_{\tilde{\mathcal{X}} \times \mathcal{Y}}$ denote a restricted distribution of $\mathbb{P}_{X,Y}$ on a subset $\tilde{\mathcal{X}} \times \mathcal{Y} \subset \mathcal{X} \times \mathcal{Y}$. Likewise, denote the standard risk w.r.t $\tilde{\mathbb{P}}_{X,Y}$ by $\tilde{\mathcal{R}}_{std}(f)$ and the robust risk w.r.t $\tilde{\mathbb{P}}_{X,Y}$ by $\tilde{\mathcal{R}}_{rob}(f)$. For a set $A$, the $\epsilon$-neighborhood in $\ell_p$-norm is defined as $B(A, \epsilon) = \{y | \|y - x\|_p \leq \epsilon, x \in A\}$, and the interior is denoted by $int(A)$. Denote the $\epsilon$-

neighborhood of the decision boundary of $f$ by $DB(f, \epsilon) = \{x \big| \exists x' \in B(x, \epsilon) \text{ s.t. } f(x) \neq f(x')\}$. Let $\hat{p}_{f,y}(x)$ denote the predicted probability of the label of $x$ being $y$ by a model $f$.

## 2 ADVERSARIAL LEARNING MAY HELP STANDARD LEARNING

In this section, we use a toy example to investigate the synergistic effect between the pursuit of robustness and natural accuracy.

**Example:** We can easily find an example of trade-off wherever an $\epsilon$-ball attack can cross the manifold boundary between classes. For example, consider a two-dimensional random vector $X = (X_1, X_2)$ on $\mathcal{X} = (0, 1) \times (0, 1)$ with a binary class $Y \sim Ber(p)$, where $p > 0.5$. Let the conditional distribution be $(X_1, X_2)|Y = 0 \sim Unif((0, \frac{1}{2}) \times (0, 1))$ and $(X_1, X_2)|Y = 1 \sim Unif((\frac{1}{2}, 1) \times (0, 1))$. Then the Bayes Rule is $f^B(x) = sign(x_1 - 0.5)$, and it is a perfect classifier, in that $\mathcal{R}_{std}(f^B) = 0$ with $\ell$ as the 0-1 loss. If $p > 0.5$, the robust classifier against $\epsilon$-ball attacks is $f_{rob}(x) = sign(x_1 - (0.5 - \epsilon))$. Its decision boundary is deviated by $-\epsilon$ from that of the Bayes rule, and this deviation costs natural accuracy by $\mathcal{R}_{std}(f_{rob}) - \mathcal{R}_{std}(f) = (1 - p)\epsilon$. However, when the data points reside in a high-dimensional space, the samples are too sparse to represent the underlying true distribution. To take this phenomenon into account, consider a distribution $\tilde{\mathbb{P}}_{X,Y} = \mathbb{P}_{X,Y}|_{\tilde{\mathcal{X}} \times \mathcal{Y}}$ on a subset $\tilde{\mathcal{X}} \times \mathcal{Y} \subset \mathcal{X} \times \mathcal{Y}$. Assume that we only observe data generated from $\tilde{\mathbb{P}}_{X,Y}$, and the training and test sets do not provide any information on $\tilde{\mathcal{X}}^c \times \mathcal{Y}$. Now by applying this support restriction to the example above, we demonstrate how adversarial learning dramatically changes from harming to benefiting natural accuracy.

**The Cheese hole distribution:** For the above example, we restrict the support on $\tilde{\mathcal{X}} = \cup_{i=1}^{3} \cup_{j=1}^{3} H_{ij}$, where $H_{ij} = ((\frac{\alpha}{2}, \frac{3\alpha}{2}) + 2\alpha(i-1)) \times ((\frac{\alpha}{2}, \frac{3\alpha}{2}) + 2\alpha(j-1))$, $\alpha = \frac{1}{6}$, and $j = 1, 2, 3$. Therefore the sampling support $\tilde{\mathcal{X}}$ comprises of nine small squares equally spaced by $\alpha$. This is illustrated in Figure 1 (a). Among all classifiers which predict the same as $f^B$ on $\tilde{\mathcal{X}}$, the worst one, denoted by $\tilde{f}^{B*}$, predicts exact opposite on $x \in \tilde{\mathcal{X}}^c$ as illustrated in Figure 1 (b). For this worst case, the true standard risk w.r.t $\mathbb{P}_{X,Y}$ is $\frac{3}{4}$, although the standard risk w.r.t. $\tilde{\mathbb{P}}_{X,Y}$ is zero.

We argue that pursuing robustness can mitigate this discrepancy. Among minimizers of $\tilde{\mathcal{R}}_{rob}$, let $\tilde{f}^*_{rob}$ be the worst classifier in that it predicts incorrectly outside $\tilde{\mathcal{X}}$ as depicted in Figure 1 (c). On $B(\tilde{\mathcal{X}}, \epsilon)$, $\tilde{f}^*_{rob}$ should correctly classify, except on $A_\epsilon = \{(x_1, x_2)|0.5 - \epsilon \leq x_1 \leq 0.5\} \cap B(\tilde{\mathcal{X}}, \epsilon)$. In this situation, the inaccuracy of $\tilde{f}^*_{rob}$ is compensated by its increased accuracy on $B(\tilde{\mathcal{X}}, \epsilon) \setminus A_\epsilon$. Moreover, the composition of $\mathcal{R}_{rob}(\tilde{f}^{B*})$ is particularly interesting, and it is easy to show that

$$\mathcal{R}_{rob}(\tilde{f}^{B*}) = \mathbb{P}\big(\tilde{f}^{B*}(Y) = X, X \in DB(\tilde{f}^{B*}, \epsilon)\big) + 3/4,$$

where 3/4 is from the standard inaccuracy on $\tilde{\mathcal{X}}^c$. The major part of $\mathcal{R}_{rob}(\tilde{f}^{B*})$ comes from the *inaccuracy* of $\tilde{f}^{B*}$ outside of the sampling support $\tilde{\mathcal{X}}$. Therefore *by simply reducing this inaccuracy on $\tilde{\mathcal{X}}^c$, we can lower the robust risk by $3/4$.* Another interesting observation is that $B_\epsilon = \{(x_1, x_2)|0.5 - \epsilon < x_1 < 0.5 + \epsilon\}$ is the only area where the Bayes rule $f^B$ has its robust risk. Note that $\mathcal{R}_{rob}(f^B) = \mathbb{P}(X \in B_\epsilon, f^B(X) = Y) = 2\epsilon$ which is greater than the optimal robust risk. The reason why $f^B$ is not most robust is simply because it is *accurate* on $B_\epsilon$. This prevents $f^B$ from being a robust function under the current adversarial robustness framework.

These observations have two implications. First, the robust pursuit can help to increase natural accuracy outside of $\tilde{\mathcal{X}}$ where the samples are poorly representative. Second, by accurately correcting the model outside of $\tilde{\mathcal{X}}$ in a standard sense, the robust risk can be significantly decreased. This is where the pursuit of robustness and accuracy coincides.

**Toward respecting the original data structure:** What if we regard the adversarial robust risk near the class border as *reasonable gullibility*? This corresponds to pursuit of robustness only against adversarial examples which do not cross between the class manifolds, i.e., *as long as this does not harm the natural accuracy on $\tilde{\mathcal{X}}$*. We call this robustness as *sensible robustness*, which respects the structure represented by data. In this example, the sensible robustness can increase both robustness and natural accuracy. Let $\tilde{f}^{s*}_{rob}$ denote the worst case sensibly robust classifier w.r.t $\tilde{\mathbb{P}}_{X,Y}$. The

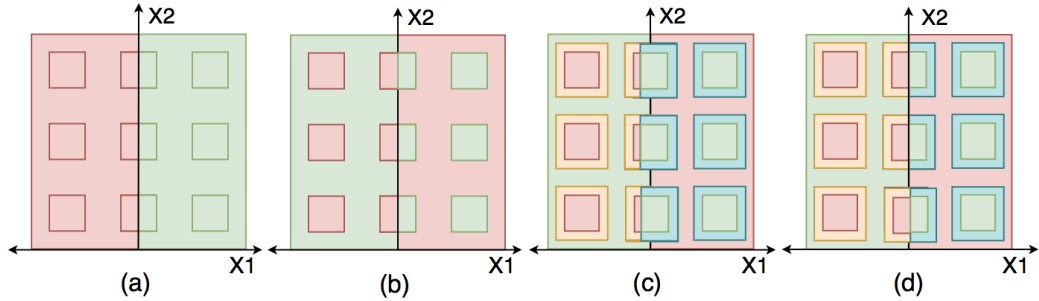

Figure 1: Cheese holes distribution. (a) The outer square is the support of the underlying true distribution $\mathbb{P}_{X,Y}$, but the sampling is restricted on the small squares $\tilde{\mathcal{X}} \times \mathcal{Y}$ with distribution $\tilde{\mathbb{P}}_{X,Y}$. (b) The worst case *naturally* optimal model w.r.t. $\tilde{\mathbb{P}}_{X,Y}$. (c) The worst case *robustly* optimal model w.r.t. $\tilde{\mathbb{P}}_{X,Y}$. (d) The worst case *sensibly* robust model w.r.t. $\tilde{\mathbb{P}}_{X,Y}$.

decision boundary of $\tilde{f}^{s*}$ should not deviate to the left on $B(\tilde{\mathcal{X}}, \epsilon)$ no matter how large $\epsilon$ is, as depicted in Figure 1 (d). Note that when $\epsilon \geq \alpha/2$, $\tilde{f}^*_{rob}$ is the unique minimizer of $\tilde{\mathcal{R}}_{rob}(f)$ as $B(\tilde{\mathcal{X}}, \epsilon)$ covers $\mathcal{X}$.

## 3 SENSIBLE ADVERSARIAL ROBUSTNESS

In this section, we introduce a sensible adversary framework. Consider a general multi-class case with the 0-1 loss, where $\mathcal{Y} = [K]$. Assume the model capacity is enough so that the Bayes rule $f^B \in \mathcal{F}$. We consider $\ell_p$-norm constrained adversarial attacks, where $p \in \{0, 1, ..., \infty\}$.

**Definition 1. (sensible adversarial example)** For a classifier $f$, let $S_{x,\epsilon}(f) = \{z \in \mathcal{X} | \|z - x\|_p \leq \epsilon, f(z) = y\}$. Then the sensible adversarial example of $(x, y)$ w.r.t $f$ is defined as

$$\tilde{x} = \begin{cases} x, & \text{if } f^B(x) \neq y \\ \arg\max_{z \in S_{x,\epsilon}(f^B)} \ell(f(z), y), & \text{otherwise.} \end{cases} \tag{2}$$

**Definition 2. (sensible robustness)** Let the sensible adversarial loss be $\ell^s_{rob,\epsilon}(f, x, y) = \ell(f(\tilde{x}), y)$ where $\tilde{x}$ is a sensible adversarial example as defined above. Then the sensible robust risk of a model $f$ is defined by

$$\mathcal{R}^s_{rob}(f) = \mathbb{E}_{\mathbb{P}_{X,Y}}\left[\ell^s_{rob,\epsilon}(f, X, Y)\right] = \mathbb{E}_{\mathbb{P}_{X,Y}}\left[\ell(f(\tilde{X}), Y)\right]. \tag{3}$$

We call its minimizer as a sensibly robust model w.r.t $\mathbb{P}_{X,Y}$ and denote by $f^s_{rob}$, i.e.,

$$f^s_{rob} = \arg\min_{f \in \mathcal{F}} \mathcal{R}^s_{rob}(f).$$

**Remark 1.** Intuitively, a sensible adversarial example is an adversarial example restricted not to cross the decision boundary of the Bayes rule. In addition, a sensible adversary does not perturb a data point that the Bayes rule incorrectly classifies. Therefore, it is natural to expect that pursuing sensible robustness would not cost natural accuracy, and the following theorem confirms it.

**Theorem 1.** Let $\mathcal{R}^*_{std}$ denote the minimum standard risk which is $\mathcal{R}_{std}(f^B)$. Then we have $\mathcal{R}^s_{rob}(f^B) = \mathcal{R}^*_{std}$. Furthermore, $f^B$ is the unique minimizer of $\mathcal{R}^s_{rob}(f)$ among $f \in \mathcal{F}$.

**Theorem 2.** Let $S_{X,\epsilon}$ be an $\epsilon$-ball centered at $X$. Then for any $f \in \mathcal{F}$ and for any set $A \subset \mathcal{X} \setminus DB(f^B, \epsilon)$,

$$\mathbb{P}\left(\exists\, x' \in S_{X,\epsilon} \text{ s.t. } f^B(x') \neq Y, X \in A\right) \leq \mathbb{P}\left(\exists\, x' \in S_{X,\epsilon} \text{ s.t. } f(x') \neq Y, X \in A\right).$$

**Remark 2.** According to the Theorem 1, a sensibly robust model w.r.t $\mathbb{P}_{X,Y}$ is the Bayes rule, i.e., $f^s_{rob} = f^B$. This sensible robustness costs adversarial robustness since $f^B$ may have a larger

---

**Algorithm 1** Sensible adversarial training for $\ell_\infty$ norm restriction

---

1: **Input:** Initialized $f = f_\theta$, $c \in (0,1)$, step number $K$, step sizes $\eta_1, \eta_2$, data $X_{adv}^{(0)} = X$
2: **repeat**
3:     **for** $i = 1, ..., m$, s.t. $f(x_{i,adv}^{(0)}) = y_i$
4:         **for** $k = 1, ..., K$
5:             $x_{i,adv}^{(k)} \leftarrow \Pi_{B(x_i,\epsilon)}(\eta_1 sign(\nabla_x \ell(f(x_{i,adv}^{(k-1)}), y_i)) + x_{i,adv}^{(k-1)})$, $\Pi$: the projection operator
6:             **if** $\ell(f, x_{i,adv}^{(k)}, y_i) > \log \frac{1}{c}$
7:                 (sensible reversion) $x_{i,adv}^{(K)} = x_{i,adv}^{(k-1)}$
8:                 **break**
9:     $\theta \leftarrow \theta - \eta_2 \sum_{i=1}^m \nabla_\theta \ell(f, x_{i,adv}^{(K)}, y_i)/m$
10: **until** training converged

---

adversarial robust risk compared with $f_{rob}$, a direct minimizer of (1). However, Theorem 2 shows $f^B$ is equally or even more robust than $f_{rob}$ except on a certain area. In particular, Theorem 2 shows that $f^B$ is the most robust model except on $DB(f^B, \epsilon)$. Therefore, $f^B$ is most robust almost everywhere if the decision boundary of $f^B$ can lie outside of $B(\mathcal{X}, \epsilon)$, so that $\mathcal{X} \setminus DB(f^B, \epsilon) = \mathcal{X}$. For example, this happens when each class has its own support apart from each other by at least $2\epsilon$.

The next theorem shows that even when we only have data from $\tilde{\mathbb{P}}_{X,Y}$ restricted on $\tilde{\mathcal{X}} \times \mathcal{Y}$, we can find $f_{rob}^s$, the optimal function w.r.t. $\mathbb{P}_{X,Y}$.

**Theorem 3.** Let $\mathcal{A}_\epsilon = \{f \in \mathcal{F} \mid \tilde{\mathbb{P}}(f(x) = f^B(x), \forall x \in S_{X,\epsilon}(f^B)) = 1\}$ and $\tilde{\mathcal{R}}_{rob}^s(f) = \mathbb{E}_{\tilde{\mathbb{P}}_{X,Y}}[\ell_{rob,\epsilon}^s(f, X, Y)]$. Then, for any $\epsilon > 0$, $\tilde{\mathcal{R}}_{rob}^s(f)$ is only minimized by any $f \in \mathcal{A}_\epsilon$. Furthermore, if $B(\tilde{\mathcal{X}}, \epsilon) \supset \mathcal{X}$, $f^B$ is the unique minimizer of $\tilde{\mathcal{R}}_{rob}^s(f)$.

**Remark 3.** It is interesting to compare our work with that of Suggala et al. (2018). We notice there are some critical differences between these two works. Given $\mathcal{Y} \in \{-1, 1\}$, Suggala et al. (2018) define an adversarial risk as

$$\mathbb{R}_{adv}(f) = \mathbb{E}[\max_{g(x)=g(x+\delta), \|\delta\| \leq \epsilon} \ell(f(x + \delta_x), g(x)) - \ell(f(x), g(x))].$$

Unlike our definition of sensible adversary, if $g$ is a Bayes rule $f^B$, their adversary tries to increase the loss w.r.t. not $y$ but a deterministic function of $x$. Therefore, for making $f^B$ being an optimal robust model, their objective should always have an additional term, e.g., $R_{nat}(f) + \lambda R_{adv}(f)$ for $0 < \lambda < \infty$ because an $f$ s.t. $f(x) \neq g(x)$ w.p. 1 can minimize $R_{adv}(f)$. Unlike their approach, our sensible adversarial risk (3) can alone be optimized making the Bayes rule as the optimal model.

## 4 ALGORITHM

A transition from theory to algorithm poses two main challenges. First, the 0-1 loss function in the theory is hard to optimize. Therefore, as a common practice, we use the cross-entropy loss. Second, $f^B$ on the entire space is practically unavailable. We note that a model that performs well on natural data can be a nice approximation of $f^B$ on the restricted support $\tilde{\mathcal{X}} \times \mathcal{Y}$. Therefore, we generalize sensible adversary in (2) to utilize a general loss function and a reference model that substitutes $f^B$.

**Definition 3.** (generalized sensible adversarial example) Consider a loss function $\ell$. For a classifier $f$ and $c \in (0, 1]$, let $S_{x,\epsilon}(f) = \{z \in \mathcal{X} \mid \|z - x\|_p \leq \epsilon, \ell(f(z), y) \leq \log \frac{1}{c}\} \cup \{x\}$. Then given a reference model $f_r$, the sensible adversarial example of $f$ for $(x, y)$ is defined as

$$\tilde{x}|_{f_r} = \begin{cases} x, & \text{if } f_r(x) \neq y \\ \arg\max_{z \in S_{x,\epsilon}(f_r)} \ell(f(z), y), & \text{otherwise.} \end{cases} \tag{4}$$

If $\ell$ is the cross-entropy loss, the condition $\ell(f_r(x), y) \leq \log \frac{1}{c}$ is equivalent to $\hat{p}_{f_r,y}(x) \geq c$. In binary case with $c = 0.5$, this requires the perturbed examples not to cross the decision boundary of $f_r$, and for general $c$, not to reach to a *vicinity* of the boundary.

Table 1: The test result on natural examples and $\ell_\infty$-attacks for CIFAR10 ($\epsilon = 8/255$). The PGD attacks are generated with 20 random restarts and counted the worst case only.

|  | NAT | CW40 | DeepFool | FGSM | LBFGS | MIFGSM | PGD100 |
|---|---|---|---|---|---|---|---|
| SENSE | 91.51 | 67.01 | 78.89 | 72.72 | 85.94 | 68.87 | 57.23 |
| TRADE | 84.92 | 62.19 | 61.38 | 61.06 | 81.58 | 57.95 | 54.72 |

Table 2: The transfer attack results between the TRADE and SENSE model on CIFAR10. The $\ell_\infty$ PGD40 and MIFGSM attacks are generated. The subscripts of the column names denote the generating model. The denominator and numerator in each cell are the number of adversarial attacks and correct predictions respectively ($\epsilon = 8/255$).

| Defence model | $PGD_{SENSE}$ | $PGD_{TRADE}$ | $MIFGSM_{SENSE}$ | $MIFGSM_{TRADE}$ |
|---|---|---|---|---|
| TRADE | 7831/10000 | 5655/10000 | 1584/3113 | 5888/10000 |
| SENSE | 6499/10000 | 6961/10000 | 6887/10000 | 2916/4112 |

In our algorithm we set $f_r$ as a naturally trained model or a current model. Given a $f_r$, the implementation of sensible adversarial attacks is straightforward. For a correctly classified natural example by $f_r$, we add perturbations in the similar way to the PGD method (Madry et al., 2017). The difference is that during the K-step of PGD iterations, once the loss of a currently generated example exceeds $\log \frac{1}{c}$, we reverse it back to the previous step and break the iteration. This requires no additional forward- or backward-propagation, compared with the PGD method. The proposed algorithm is in Algorithm 1 for the $\ell_\infty$-norm and in Algorithm 2 in Appendix C for the other $\ell_p$-norms.

The definition of the sensible adversary in (4) divides the data into three subsets: $A_{f_r} = \{x | f_r(x) \neq y \text{ or } S_{x,\epsilon}(f_r) = \{x\}\}$, $B_{f_r} = \{x | S_{x,\epsilon}(f_r) \neq S_{x,\epsilon}, S_{x,\epsilon}(f_r) \neq \{x\}\}$, and $C_{f_r} = \mathcal{X} \setminus (A_f \cup B_f)$. Therefore, our algorithms generate a sensible adversarial example $\tilde{x}^s |_{f_r=f}$ in three different ways: (1) $\tilde{x}^s = x$ for $x \in A_f$, (2) $\tilde{x}^s \neq \tilde{x}^p$ for $x \in B_f$, and (3) $\tilde{x}^s = \tilde{x}^p$ for $x \in C_f$, where $\tilde{x}^p$ is a full K-step PGD attack. Therefore, sensible adversarial training inherently involves a set selection mechanism. The sensible adversarial loss $\ell^s(f, x, y)$ can be written as

$$\ell(f, x, y)1_{x \in A_f} + \ell(f, \tilde{x}^s, y)1_{x \in B_f} + \ell(f, \tilde{x}^p, y)1_{x \in C_f}$$
$$= \ell(f, x, y)1_{\ell(f,x,y) > \log \frac{1}{c}} + \ell(f, \tilde{x}^s, y)1_{\tilde{x}^p \neq \tilde{x}^s, \ell(f,x,,y) \leq \log \frac{1}{c}} + \ell(f, \tilde{x}^p, y)1_{\ell(f,\tilde{x}^p,y) \leq \log \frac{1}{c}}. \quad (5)$$

When $c \geq 0.5$, this optimization problem has a strong motivation to increase $|C_f|$, i.e., the number of sensible adversarial attacks that are identical to full PGD attacks. Because $A_f = \{x | \ell(f(x), y) > \log \frac{1}{c}\}$ for $c \geq 0.5$, the loss on the natural stage $A_f$ is always greater than the loss of the sensibly reversed stage $B_f$ and the full PGD stage $C_f$. Furthermore, the loss for $x \in B_f$ is always approximately $\log \frac{1}{c}$ by adaptive sensible perturbations. Therefore, paradoxically, the smallest loss is only achievable by full PGD examples. In other words, sensible adversarial training penalizes when $x \notin C_f$. We note that the optimization problem has a smooth landscape; Although the data points may jump around between $A_f$, $B_f$, and $C_f$, there is no obvious discontinuity in the loss. Therefore, during the training, $\ell^s(f, x, y)$ smoothly slides down, making both $\ell(f, x, y)$ and $\ell(f, \tilde{x}^p, y)$ smaller than $\log \frac{1}{c}$. This enables to learn a naturally and adversarially accurate model. More discussion on the stability of our algorithms are presented in Appendix D.

## 5 EXPERIMENTS

### 5.1 EXPERIMENT 1: ROBUSTNESS AGAINST $\epsilon$-BALL ATTACKS

**CIFAR10** We train WideResNet-34-10 (He et al., 2016) with sensible adversarial examples of $\epsilon = 8/255$ and $c = 0.7$ with the step number $K = 10$ and size $\eta_1 = \epsilon/5$. We attack our model with various white-box attacks with perturbations restricted to $\ell_\infty$ with $\epsilon = 8/255$. We compare the performance with TRADE using the same architecture (Zhang et al., 2019), which is know to be robust and accurate. The result is in Table 1. Our model achieves 91.51% natural accuracy. This is 3.7% drop in natural accuracy from 95.2%, which is an accuracy that a naturally trained model can achieve (Madry et al., 2017). With this architecture, Madry et al. (2017) achieve 47.04% robust

Table 3: MNIST: test results of our models on natural examples and $\ell_\infty$ based attacks ($\epsilon = 0.3$).

| Defence model | Natural | PGD500 | C&W40 |
|---|---|---|---|
| SENSE | 99.51 | 91.74 | 96.02 |
| TRADE | 99.48 | 93.30 | 96.90 |

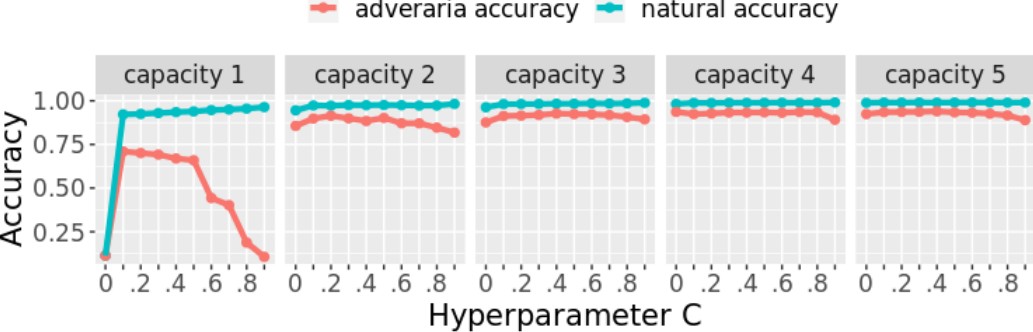

Figure 2: The natural and robust accuracies of our models for the varying parameter $c$.

accuracy against PGD20 attacks and 87.3% natural accuracy. As a black-box attack, we try transfer attacks between the TRADE and SENSE model by the $\ell_\infty$ based PGD and MIFGSM, which are known to be effective for transfer attack ($\epsilon = 8/255$). We obtain adversarial examples by applying PGD and MIFGSM on a generating model, and then use the examples to attack a defense model. The result is in Table 2. Overall we observe that our model outperforms both the TRADE and Madry model. In particular, sensible adversary achieves more than $55\%$ of robust accuracy against PGD attacks of $\epsilon = 8/255$. This performance is consistent to the test margin distribution in Figure 8, and this is discussed in detail in Appendix E. For PGD100, we conduct random 20 restarts and count only the worst case. The step number 100 and step size 2/255 of the PGD attacks are justified by the plot in Figure 11 in Appendix H.

**MNIST**   We consider a CNN model with three convolutional layers followed by a fully connected linear layer, which is the same architecture in (Zhang et al., 2019). We train an MNIST model with sensible adversarial examples of $\epsilon = 0.3$ and $c = 0.5$ with the step number $K = 10$ and size $\eta_1 = 0.05$. The robust test result in Table 3 shows the comparable performance of our model. The step size of the PGD500 attack is 0.01, and the serenity check on the PGD attack is in Appendix H. We conduct 100 random restarts and count only the worst case for each test example.

## 5.2   EXPERIMENT 2: SENSITIVITY ANALYSIS

**MNIST:** We perform the sensibility analysis to understand the effect of $c$ and the model capacity for two reasons. First, the sensible training prevents full PGD perturbations on an example until the loss on it becomes less than $\log \frac{1}{c}$, which could hardly happen for a model with a small capacity. Second, as Madry et al. (2017) point out, when the model capacity is insufficient for adversarial learning, the model collapses to a constant function. We are interested in a range of $c$ that keeps sensible learning from collapsing. Therefore, we consider a sequence of CNNs of the increasing number of kernels similar to Madry et al. (2017), where we denote the capacity by $q \in \{1, 2, 3, 4, 5\}$. The details of the model architectures are presented in Appendix H. When we train them with natural examples, the networks of capacity 1 and 2 achieve about 95% and 97% accuracy, whereas the networks of the other capacities achieve more than 99%. When trained with regular PGD examples, the networks with capacity 1,2, and 3 collapse. Therefore, capacity 3 is enough only for natural learning, and capacities 1 and 2 are possibly insufficient even for natural learning.

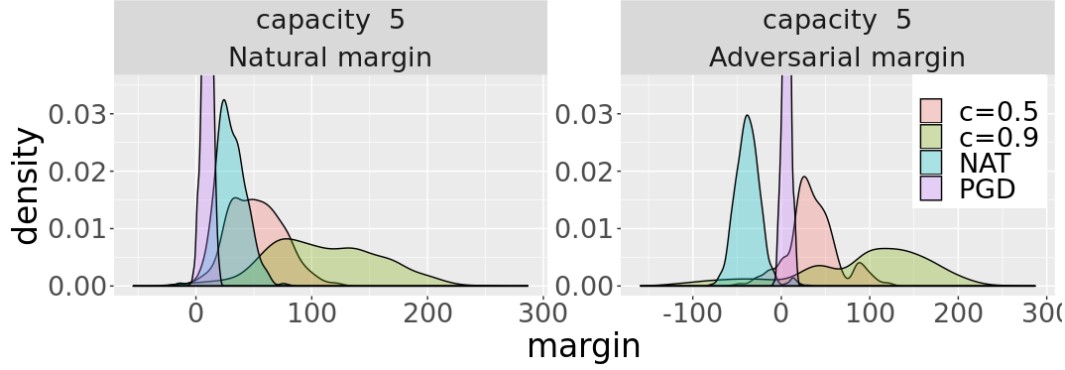

Figure 3: The prediction margins at convergence of capacity 5 on the test set. The natural margin of a model $f$ at $(x, y)$ is $\log \hat{p}_{f,y}(x) - \max_{y' \neq y} \log \hat{p}_{f,y'}(x)$. The adversarial margin is calculated by $\log \hat{p}_{f,y}(\tilde{x}^p) - \max_{y' \neq y} \log \hat{p}_{f,y'}(\tilde{x}^p)$, where $\tilde{x}^p$ is a full PGD attack.

For each capacity $q$, we train the models with sensible adversarial examples on MNIST with the hyperparameters $c \in \{0, 0.1, \cdots, 0.9, 1\}$. Note that sensible adversarial training is identical to natural training when $c = 1$, and to adversarial learning without any perturbations for incorrectly classified natural examples when $c = 0$. Figure 2 shows the natural and robust accuracy against PGD-40 attacks with varying $c$ for each capacity. In general, the natural accuracy tends to increase as $c$ increases while robustness decreases. For capacities 3, 4 and 5, the accuracies are almost insensitive to varying $c$. On the other hand, the tendency is most distinct in capacity 1. In this case, the network obtains best robustness when $c = 0.5$, which is the least loss bound among $c \geq 0.5$, the range of $c$ with stable learning property. Even for capacity 1, however, the sensible learning does not collapse except when $c = 0$. We also see that $c = 0.5$ is least sensitive to the model capacity.

Note that capacities 2 and 3 do not collapse even with $c = 0$. When $c = 0$, the only difference of the sensible adversarial learning from the regular PGD training is that the sensible learning requires robustness only for the correctly classified natural examples. When $c = 0$, at the convergence of the models of capacities 1 and 2, about 5% of the data points are allowed to be free from the perturbation. By paying only the $5\%$ of robust training accuracy, the sensible learning avoids collapsing and obtains about 80% of robust accuracy and 90% of natural accuracy.

Intuitively, an insufficient model capacity or locally close class manifolds can make a virtual decision boundary that is inevitable to keep a nice natural performance. In the algorithm, the sensible reversion prevents adversarial examples from crossing this boundary. This effectively reduces the requirement of the model capacity posed by the regular adversarial learning. The sensible reversion also allows a robust model to have larger margins than the PGD trained model for the majority of the dataset as in Figure 3. The PGD trained model has majority adversarial margins as positive. Instead, it has much smaller natural margins than the naturally trained model. The SENSE model with $c = 0.5$ has comparably large adversarial and natural margins. Instead, the number of data points with negative adversarial margins is larger than that of the PGD trained model. As $c$ increases to 0.9, this model has much smaller and more negative adversarial margins. On the other hand, its two types of margins are generally even larger. This phenomenon is consistent to the decreasing robustness in Figure 2 for capacity 5. In general, for a fixed capacity, increasing $c$ increases the natural and adversarial margins of the majority of the data, while it also increases the portion of data of negative adversarial margins.

The cost for this highly confident prediction is the robustness near the decision boundary at convergence, i.e., the portion of the data points in the natural and sensibly reversed stage. When we consider the margin on the training set, we find that there is a linear relationship between the portion of the data points having negative adversarial margins and the test accuracy. In practice, such a portion can be an indicator about the robustness of the model or the sufficiency of the model capacity, without directly testing the model.

Table 4: The sensitivity of $c$ on CIFAR models of WideResNet

| CIFAR WideResNet | c=0.0 | c=0.3 | c=0.5 | c=0.6 | c=0.7 | c=0.8 |
|---|---|---|---|---|---|---|
| Natural data | 82.88 | 86.76 | 90.42 | 90.87 | 91.51 | **92.35** |
| PGD100 | 43.70 | 46.90 | 50.95 | 55.90 | **57.80** | 55.60 |

Table 5: The sensitivity of $c$ on CIFAR models with CNNs

| CIFAR CNNs | c=0.1 | c=0.5 | c=0.9 | natural training |
|---|---|---|---|---|
| NAT | 66.26 | 75.70 | 82.02 | 80.85 |
| PGD40 | 26.67 | 20.26 | 3.95 | 0.00 |

**CIFAR:** We train train WideResNet-34-10 with several different $c$ values. Except that for $c = 0.8$ we stopped learning at 120 epoch, other models are trained for 300 epochs. We report the adversarial accuracy against and PGD100 attacks with a step size $\eta_1 = 2/255$ for the first 2000 test examples, with random 20 restarts. The results are in Table 4.

Interestingly, while the natural accuracy positively correlated to the $c$ value, the robustness does not show clear negative correlation to $c$. Rather, as $c$ become closer to 0.7, more robust result the model shows. As we see that $c = 0.8$ has better robustness than $c \leq 0.5$, in CIFAR, the main reason of the observed trade-off could attribute to the influential adversarial perturbations.

For the CIFAR dataset, we also try to train a robust model with a small capacity. We intentionally choose the CNNs model that is used in Experiment 1 for MNIST. This model can be not enough for adversarial training on CIFAR. We report the results in Table 5. Given the serious lack of model capacity, the decrease in natural accuracy of SENSE models compared with the naturally trained model is not serious, while achieving better robust accuracy numbers. Note that none of our models collapse, and therefore the possibility of sensible adversarial training is not sensitive to the choice of $c$ even in the lack of model capacity.

## 6    CONCLUDING REMARKS

In this paper, we proposed a sensible adversary which is useful for learning a defense model, keeping a high natural accuracy simultaneously. We theoretically establish that the Bayes rule is most robust under the framework of sensible adversarial learning. Our learning algorithm is efficient and stable, and not sensitive to the choice of the main hyperparameter $c$. Also, $c$ has a clear meaning as the lowest prediction-probability bound. Our empirical experiments yield state-of-the-art results of adversarial learning on the CIFAR10 and MNIST datasets. In addition, we showed that the sensible approach can effectively deal with the lack of model capacity. This is because by paying a robust accuracy on a certain area, the algorithm protects the model from being collapsed by influential adversarial examples. Furthermore, the sensible adversarial learning trains a model to have high prediction margins on both natural and adversarial examples.

We now mention several future directions for research on sensible adversary. One remaining theoretical problem is to develop generalization error bounds for sensible adversary learning, so that we can theoretically justify our empirical performance. In fact, in this work, we did not tackle the lack of sample size. In particular, as our algorithm tends to produce a model with large natural and adversarial prediction margins, in the lack of sample size, it is not clear if this large margins are always beneficial. Therefore, there is much remaining work to be done to theoretically understand the high margin tendency of the models trained with sensible adversarial examples with relation to generalization.

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

# Appendices

## A ADDITIONAL RELATED WORK

Bubeck et al. (2018) conjecture that learning a robust model is information theoretically possible but computationally intractable. They introduce an example which is not robustly learnable in polynomial time. Our view is consistent with Bubeck et al. (2018) in that a robust model exists, but we search for an efficient algorithm to estimate a robust model in a reasonable time. Su et al. (2018) compared various naturally trained models on ImageNet, and found that the trade-off varies among different model architectures. Also, they empirically discovered that more accurate models tend to be less robust when the models are trained with natural examples. Stutz et al. (2019) demonstrate that given a large training set, adversarial training can produce a robust model that is as accurate as a naturally trained model. Our work to maximize the synergistic effect between natural and adversarial accuracy is consistent to their demonstration. Also, they show that most of PGD attacks are off-manifold of the original data, and by on-manifold adversarial training, the natural accuracy can be improved. If we see the restricted space $\tilde{\mathcal{X}}$ as the data manifold, and $\tilde{\mathcal{X}}^c$ as off-manifold, our sensible framework aligns with their view, given the data manifolds of different classes are separated by at least $2\epsilon$. Kurakin et al. (2016b) suggest adversarial learning that trains with data randomly divided into two parts, a natural and adversarial set. We divide data in a *data adaptive* way into three parts including a *sensibly reversed* adversarial set. The relationship between their approach and our algorithm is discussed more in Appendix F.

## B PROOFS OF THEOREMS

To save the space, we use $\eta(k|x)$ to denote $\mathbb{P}(Y = k|X = x)$.

**Theorem 1.** (Restated) Let $\mathcal{R}^*_{std}$ denote the minimum standard risk which is $\mathcal{R}_{std}(f^B)$. Then we have $\mathcal{R}^s_{rob}(f^B) = \mathcal{R}^*_{std}$. Furthermore, $f^B$ is the unique minimizer of $\mathcal{R}^s_{rob}(f)$ among $f \in \mathcal{F}$.

*Proof.*

$$
\begin{aligned}
\mathcal{R}^s_{rob}(f^B) &= \mathbb{P}(f^B(\tilde{X}) \neq Y) \\
&= \mathbb{P}(f^B(X) \neq Y) + \mathbb{P}(f^B(X) = Y, \text{and } \exists x' \in S_{X,\epsilon}(f^B) \ s.t. \ f^B(x') \neq Y) \\
&= \mathbb{P}(f^B(X) \neq Y) + 0, \quad \text{by the definition of } S_{X,\epsilon}(f^B) \\
&= \mathcal{R}_{std}(f^B) = \mathcal{R}^*_{std}.
\end{aligned}
$$

It is obvious that $f^B$ is a minimizer of $\mathcal{R}^s_{rob}(f)$ because $\mathcal{R}^*_{std}$ is a lower bound of $\mathcal{R}^s_{rob}(f)$ for any $f \in \mathcal{F}$.

Now we show $f^B$ is the unique minimizer of $\mathcal{R}^s_{rob}(f)$.

$$
\begin{aligned}
\mathcal{R}^s_{rob}(f) &= \mathbb{P}(f^B(X) \neq Y, f(X) \neq Y) + \mathbb{P}(f^B(X) = Y, \exists x' \in S_{X,\epsilon}(f^B) \ s.t. \ f(x') \neq Y) \\
&= \sum_{k=1}^{K} \mathbb{P}(f^B(X) \neq k, f(X) \neq k, Y = k) + \mathbb{P}(f^B(X) = k, \exists x' \in S_{X,\epsilon}(f^B) \ s.t. \ f(x') \neq k, Y = k) \\
&= \sum_{k=1}^{K} \int_{\mathcal{X}} \mathbb{P}(f^B(x) \neq k, f(x) \neq k, Y = k | X = x) \\
&\quad + \mathbb{P}(f^B(x) = k, \exists x' \in S_{x,\epsilon}(f^B) \ s.t. \ f(x') \neq k, Y = k | X = x) d\mathbb{P}(x) \\
&= \sum_{k=1}^{K} \int_{\mathcal{X}} 1_{f^B(x) \neq k, f(x) \neq k} \eta(k|x) + 1_{f^B(x) = k, \exists x' \in S_{x,\epsilon}(f^B) \ s.t. \ f(x') \neq k} \eta(k|x) d\mathbb{P}(x) \\
&= \mathcal{R}_{std}(f) + \sum_{k=1}^{K} \int_{\mathcal{X}} 1_{f^B(x) = k} (1_{\exists x' \in S_{x,\epsilon}(f^B) \ s.t. \ f(x') \neq k} - 1_{f(x) \neq k}) \eta(k|x) d\mathbb{P}(x) \quad (6)
\end{aligned}
$$

The last equality is by $1_{f^B(x) \neq k, f(x) \neq k} = (1 - 1_{f^B(x) = k}) 1_{f(x) \neq k}$.

The first term $\mathcal{R}_{std}(f)$ is uniquely minimized by the Bayes rule $f^B$. The second term is always non-negative, and is zero when $f = f^B$. Therefore, $\mathcal{R}^s_{rob}(f)$ is uniquely minimized by $f^B$.

$\square$

**Theorem 2.** (Restated) Let $S_{X,\epsilon}$ be an $\epsilon$-ball centered at $X$. Then for any $f \in \mathcal{F}$ and for any set $A \subset \mathcal{X} \setminus DB(f^B, \epsilon)$,

$$\mathbb{P}\Big(\exists\, x' \in S_{X,\epsilon} \text{ s.t. } f^B(x') \neq Y, X \in A\Big) \leq \mathbb{P}\Big(\exists\, x' \in S_{X,\epsilon} \text{ s.t. } f(x') \neq Y, X \in A\Big).$$

*Proof.* For any set $A \subset \mathcal{X} \setminus DB(f^B, \epsilon)$, $\mathbb{P}(\exists x' \in S_{X,\epsilon} \text{ s.t. } f^B(x') \neq Y, X \in A) = \mathbb{P}(f^B(X) \neq Y, X \in A)$. Note that on any subset $B \subset \mathcal{X}$, the Bayes rule has the least error probability. Therefore, $\mathbb{P}(f^B(X) \neq Y, X \in A) \leq \mathbb{P}(f(X) \neq Y, X \in A) \leq \mathbb{P}(\exists x' \in S_{X,\epsilon} \text{ s.t. } f(x') \neq Y, X \in A)$. The last inequality is trivial because if $f(X) \neq Y \Rightarrow \exists x' \in S_{X,\epsilon} \text{ s.t. } f(x') \neq Y$. $\square$

**Theorem 3.** (Restated) Let $\mathcal{A}_\epsilon = \{f \in \mathcal{F} | \tilde{\mathbb{P}}(f(x) = f^B(x), \forall x \in S_{X,\epsilon}(f^B)) = 1\}$ and $\tilde{\mathcal{R}}^s_{rob}(f) = \mathbb{E}_{\tilde{\mathbb{P}}_{X,Y}}[\ell^s_{rob,\epsilon}(f, X, Y)]$. Then, for any $\epsilon > 0$, $\tilde{\mathcal{R}}^s_{rob}(f)$ is only minimized by any $f \in \mathcal{A}_\epsilon$. Furthermore, if $B(\tilde{\mathcal{X}}, \epsilon) \supset \mathcal{X}$, $f^B$ is the unique minimizer of $\tilde{\mathcal{R}}^s_{rob}(f)$.

*Proof.* The sensible risk of $f$ w.r.t. the restricted distribution corresponding to (6) is

$$\tilde{\mathcal{R}}^s_{rob}(f) = \tilde{\mathcal{R}}_{std}(f) + \sum_{k=1}^{K} \int_{\tilde{\mathcal{X}}} 1_{f^B(x)=k}(1_{\exists x' \in S_{x,\epsilon}(f^B) \text{ s.t. } f(x') \neq k} - 1_{f(x) \neq k}) \eta(k|x) d\tilde{\mathbb{P}}(x) \tag{7}$$

$$= \tilde{\mathcal{R}}_{std}(f) + \int_{\tilde{\mathcal{X}}} \sum_{k=1}^{K} 1_{f^B(x)=k}(1_{\exists x' \in S_{x,\epsilon}(f^B) \text{ s.t. } f(x') \neq f^B(x)} - 1_{f(x) \neq f^B(x)}) \eta(f^B(x)|x) d\tilde{\mathbb{P}}(x)$$

$$= \tilde{\mathcal{R}}_{std}(f) + \int_{\tilde{\mathcal{X}}} (1_{\exists x' \in S_{x,\epsilon}(f^B) \text{ s.t. } f(x') \neq f^B(x)} - 1_{f(x) \neq f^B(x)}) \eta(f^B(x)|x) d\tilde{\mathbb{P}}(x) \tag{8}$$

because $\tilde{\mathbb{P}}(Y = k|X = x) = \mathbb{P}(Y = k|X = x)$ for $x \in \tilde{\mathcal{X}}$. The minimum of $\tilde{\mathcal{R}}^s_{rob}(f)$ is achieved by $f^B$ with the minimum value $\tilde{\mathcal{R}}^* = \tilde{\mathcal{R}}_{std}(f^B) + 0$. Therefore, any function $f$ that $\tilde{\mathcal{R}}_{std}(f) > \tilde{\mathcal{R}}^*$ cannot achieve the minimum of $\tilde{\mathcal{R}}^s_{rob}(f)$ because the term $(1_{\exists x' \in S_{x,\epsilon}(f^B) \text{ s.t. } f(x') \neq f^B(x)} - 1_{f(x) \neq f^B(x)})$ in (8) is always non-negative. Therefore, only functions in $\mathcal{A} = \{f \in \mathcal{F} | \tilde{\mathbb{P}}(f(X) = f^B(X)) = 1\}$ need to be considered as possible minimizers of $\tilde{\mathcal{R}}^s_{rob}(f)$.

Note that $\mathcal{A}_\epsilon \subset \mathcal{A}$. By the definition, we know that $f \in \mathcal{A} \setminus \mathcal{A}_\epsilon$ if and only if

$$i) \tilde{\mathbb{P}}\big(f(X) = f^B(X)\big) = 1$$
$$ii) \tilde{\mathbb{P}}\big(f(x) = f^B(x), \forall x \in S_{X,\epsilon}(f^B)\big) < 1$$

Therefore, for $f \in \mathcal{A} \setminus \mathcal{A}_\epsilon$, $\exists A \subset \tilde{\mathcal{X}}$ s.t. $\tilde{\mathbb{P}}(X \in A) > 0$ and $\exists x' \in S_{x,\epsilon}(f^B)$ for $\forall x \in A$ s.t. $f(x') \neq f^B(x')$. For this $f$, the equation in (8) can be written as $\tilde{\mathcal{R}}^s_{rob}(f) = \tilde{\mathcal{R}}^* + \alpha$ for some $\alpha \geq 0$. Now we show that $\alpha > 0$.

Note that by the definition of the Bayes rule, $\mathbb{P}(y = f^B(x)|X = x) \geq \frac{1}{K}$. Otherwise, $1 = \sum_{k=1}^{K} \mathbb{P}(y = k|X = x) \leq K\mathbb{P}(y = f^B(x)|X = x) < 1$, which is contradict. Then, for the $f \in \mathcal{A} \setminus \mathcal{A}_\epsilon$ and $A \subset \tilde{\mathcal{X}}$ that are described above,

$$\int (1_{\exists x' \in S_{x,\epsilon}(f^B) \text{ s.t. } f(x') \neq f^B(x)} - 1_{f(x) \neq f^B(x)}) \eta(f^B(x)|x) d\tilde{\mathbb{P}}(x)$$

$$\geq \frac{1}{K} \int_{\tilde{\mathcal{X}}} (1_{\exists x' \in S_{x,\epsilon}(f^B) \text{ s.t. } f(x') \neq f^B(x)} - 1_{f(x) \neq f^B(x)}) d\tilde{\mathbb{P}}(x)$$

$$= \frac{1}{K} \int_{\tilde{\mathcal{X}}} (1_{\exists x' \in S_{x,\epsilon}(f^B) \text{ s.t. } f(x') \neq f^B(x)}) d\mathbb{P}(x) \quad \text{by } f \in \mathcal{A}$$

$$\geq \frac{1}{K} \int_A (1_{\exists x' \in S_{x,\epsilon}(f^B) \text{ s.t. } f(x') \neq f^B(x)}) d\mathbb{P}(x) = \frac{\mathbb{P}(A)}{K} > 0.$$

Therefore, $\alpha > 0$. Note that for any $f \in \mathcal{A}_\epsilon$, the second term in (8) is zero by the definition of $\mathcal{A}_\epsilon$. Therefore, first result of the theorem is proved. Furthermore, for $\epsilon$ such that $B(\tilde{\mathcal{X}}, \epsilon) \supset \mathcal{X}$, $\mathcal{A}_\epsilon = \{f^B\}$. Therefore, when $B(\tilde{\mathcal{X}}, \epsilon) \supset \mathcal{X}$, $f^B$ is the unique minimizer of $\tilde{\mathcal{R}}^s_{rob}(f)$. $\square$

**Theorem 4.** Let $\mathcal{A} = \{f \in \mathcal{F} | \tilde{\mathbb{P}}(f(X) = f^B(X)) = 1\}$, and take a reference function $f_r \in \mathcal{A}$. Consider extended sensible adversarial examples, with $c < 1$ and $\ell$ as the 0-1 loss. Then, for any $\epsilon > 0$, $\tilde{\mathcal{R}}^s_{rob}(f|f_r) = \mathbb{E}_{\tilde{\mathbb{P}}_{X,Y}}[\ell(f(\tilde{X}|f_r), Y)]$ is uniquely minimized by $f_r$.

Theorem 4 says that for $f_r \in \mathcal{A}$, which behaves as the Bayes rule $f^B$ on the restricted support, the corresponding sensible adversarial risk $E_{\tilde{\mathbb{P}}_{X,Y}}[\ell(f(\tilde{X}|f_r), Y)]$ is minimized only by $f_r$. This implies that if we do not have any information about the Bayes rule on $\tilde{\mathcal{X}}^c$, the sensibly optimal model w.r.t. $\tilde{\mathbb{P}}_{X,Y}$ can be arbitrary on $\tilde{\mathcal{X}}^c$ although this optimal model is the Bayes rule on $\tilde{\mathcal{X}}$. Our algorithm deals with this arbitrariness by searching for a better reference function in each iteration. As a current model is used as a reference function, i.e., the estimation of the defense model and the reference model is identical, the algorithm essentially pursues sensibleness on $\tilde{\mathcal{X}}$ and robustness on $\tilde{\mathcal{X}}^c$ of the trained model. Note that on $\tilde{\mathcal{X}}$, sensibleness a sufficient condition for natural accuracy.

*Proof.* By using the same way to derive (6) and (8) and noting that $f_r(x) = f^B(x)$ on $\tilde{\mathcal{X}}$, we get

$$\tilde{\mathcal{R}}^s_{rob}(f|f_r) = \tilde{\mathcal{R}}_{std}(f) + \sum_{k=1}^{K} \int_{\tilde{\mathcal{X}}} 1_{f^B(x)=k}(1_{\exists x' \in S_{x,\epsilon}(f_r) \text{ s.t. } f(x') \neq k} - 1_{f(x) \neq k})\eta(k|x)d\tilde{\mathbb{P}}(x)$$

$$= \tilde{\mathcal{R}}_{std}(f) + \int_{\tilde{\mathcal{X}}} (1_{\exists x' \in S_{x,\epsilon}(f_r) \text{ s.t. } f(x') \neq f^B(x)} - 1_{f(x) \neq f^B(x)})\eta(f^B(x)|x)d\tilde{\mathbb{P}}(x) \quad (9)$$

Because $f_r(x) = f^B(x)$ on $\tilde{\mathcal{X}}$, $\tilde{\mathcal{R}}_{std}(f_r) = \tilde{\mathcal{R}}_{std}(f^B)$. This implies $f_r$ minimizes $\tilde{\mathcal{R}}^s_{rob}(f|f_r)$ because $\tilde{\mathcal{R}}_{std}(f^B)$ is the minimum of $\tilde{\mathcal{R}}_{std}(f)$ for all $f \in \mathcal{F}$ and the second term in (9) is non-negative. This implies any function $f$ s.t. $\tilde{\mathcal{R}}_{std}(f) > \tilde{\mathcal{R}}_{std}(f^B)$ cannot achieve the minimum of $\tilde{\mathcal{R}}^s_{rob}(f|f_r)$. Therefore, as a minimizer of $\tilde{\mathcal{R}}^s_{rob}(f|f_r)$, we only need to consider $f \in \mathcal{A}$. Note that $1_{f(x) \neq f^B(x)} = 0$ for $f \in \mathcal{A}$ on $\tilde{\mathcal{X}}$. Therefore, by letting $\tilde{\mathcal{R}}^*_{std}$ be $\tilde{\mathcal{R}}_{std}(f^B)$, the equation in (9) can be written as

$$\tilde{\mathcal{R}}^s_{rob}(f|f_r) = \tilde{\mathcal{R}}^*_{std} + \int_{\tilde{\mathcal{X}}} 1_{\exists x' \in S_{x,\epsilon}(f_r) \text{ s.t. } f(x') \neq f^B(x)}\eta(f^B(x)|x)d\tilde{\mathbb{P}}(x)$$

$$= \tilde{\mathcal{R}}^*_{std} + \int_{\tilde{\mathcal{X}}} 1_{\exists x' \in S_{x,\epsilon}(f_r) \text{ s.t. } f(x') \neq f_r(x)}\eta(f^B(x)|x)d\tilde{\mathbb{P}}(x) \quad (10)$$

Note that $\eta(f^B(x)|x)$ is positive on $\tilde{\mathcal{X}}$. Therefore, for $f \in \mathcal{A}$ to minimize (10), it must satisfy that $\tilde{\mathbb{P}}(\exists x' \in S_{X,\epsilon}(f_r) \text{ s.t. } f(x') \neq f_r(x')) = 0$. This essentially says $f$ should be $f_r$.

$\square$

## C  ALGORITHM

---

**Algorithm 2** Sensible adversarial training for $\ell_p$ norm restriction

---

    **Input:** Initialized $f = f_\theta$, $c \in (0, 1)$, step number and sizes $K, \eta_1, \eta_2$, data $X^{(0)}_{adv} = X$

2: **repeat**

      **for** $i = 1, ..., m$, s.t. $f(x^{(0)}_{i,adv}) = y_i$

4:        **for** $k = 1, ..., K$

            $x^{(k)}_{i,adv} \leftarrow \Pi_{B_p(x_i, \epsilon)}(\eta_1 \frac{\nabla_x \ell(f(x^{(k-1)}_{i,adv}), y_i)}{\|\nabla_x \ell(f(x^{(k-1)}_{i,adv}), y_i)\|_p} + x^{(k-1)}_{i,adv})$, $\Pi$: the projection operator

6:           **if** $\ell(f, x^{(k)}_{i,adv}, y_i) > \log \frac{1}{c}$

           (sensible reversion) $x^{(K)}_{i,adv} = x^{(k-1)}_{i,adv}$

8:           **break**

      $\theta \leftarrow \theta - \eta_2 \sum_{i=1}^{m} \nabla_\theta \ell(f, x^{(K)}_{i,adv}, y_i)/m$

10: **until** training converged

---

Algorithm 2 is a straightforward extension of Algorithm 1 from $\ell_\infty$ norm to $\ell_p$ norm. In addition, we present another way to realize the sensible reversion. After stepping back when the loss exceeds the threshold, we can add a random noise as the next step instead of just breaking the iteration as in Algorithm 1 and 2. Considering the nature of (mini-) batch leaning, this random noise does not add much computational cost because until every loss exceeds the threshold, the for loop would keep calculating the forward and backward loop with fixed perturbation for the reversed examples. This algorithm for $\ell_\infty$ norm is presented in Algorithm 3, of which the objective function is also (5). Algorithm 3 also reverses the adversarial example when the loss is greater than $\log \frac{1}{c}$. The only difference is that instead of breaking the iteration after the reversion, it considers a random noise as the next step. This can lead to more effective search for the local loss maximizer in the ball $_{x,\epsilon}(f)$ in definition 4. The extension of Algorithm 3 to $\ell_p$ norm is straight forward, and thus omitted.

---

**Algorithm 3** Sensible adversarial training for $\ell_\infty$ norm restriction

---

    **Input:** Initialized $f = f_\theta$, $c \in (0,1)$, step number $K$, step sizes $\eta_1, \eta_2$, data $X_{adv}^{(0)} = X$
    **repeat**
3:    **for** $i = 1, ..., m$, s.t. $f(x_{i,adv}^{(0)}) = y_i$
        **for** $k = 1, ..., K$
           $\tilde{x}_i \leftarrow \Pi_{B(x_i,\epsilon)}(step_i^{(k)} + x_{i,adv}^{(k-1)})$, $\Pi$: the projection operator
6:        **if** $\ell(f, \tilde{x}_i, y_i) \leq \log \frac{1}{c}$
           $x_{i,adv}^{(k)} = \tilde{x}_i$
           $step_i^{(k+1)} = \eta_1 sign(\nabla_x \ell(f(x_{i,adv}^{(k)}), y_i))$
9:        **else**
           $x_{i,adv}^{(k)} = x_{i,adv}^{(k-1)}$
         $step_i^{(k+1)} = $ random noise
12:    $\theta \leftarrow \theta - \eta_2 \sum_{i=1}^m \nabla_\theta \ell(f, x_{i,adv}^{(K)}, y_i)/m$
    **until** training converged

---

# D  THE LANDSCAPE FOR SENSIBLE ROBUST OPTIMIZATION

Note that when the original examples are incorrectly classified, full PGD examples can be very influential in regular adversarial training. If we do not add any perturbations on these potentially influential examples and add full PGD perturbations on the other examples, the resultant examples are equivalent to the sensible adversarial examples when $c = 0$. Experiment 1 shows this small change keeps the models from collapsing, demonstrating how influential the PGD perturbations on the incorrectly classified natural examples. However, an adversarial training with the sensible adversarial examples with $c = 0$, the empirical loss can largely fluctuate during the training; Once an incorrectly classified natural example becomes correctly classified, its full PGD attack can pose a sudden large gradient for the model update. Then, when it is incorrectly classified again, the loss on it suddenly reduces to the natural loss that is distinctly smaller than the large loss value on its full PGD attack. Therefore, whenever an example changes its state between a correctly and incorrectly classified example, i.e., the full PGD and natural stage, the corresponding loss can fluctuate making the learning unstable.

However, if we add sensible reversion step, particularly if $c \geq 0.5$, this fluctuation does not occur. Between the two stages, the sensibly reversed stage provides a kind of cushion between the natural and full PGD stage, preventing the sudden change in the loss value as described. Until $S_{x,\epsilon}(f) = S_{x,\epsilon}$, the sensibly adversarial perturbation for $x$ is adapted to make the loss of the current function approximately equal to $\log \frac{1}{c}$. As the abstraction in Figure 4, it is like to have a virtually extended area at $\hat{p}_{f,y}(x) = c$ on the loss function that is locally flat on $\{\tilde{x}^s | S_{x,\epsilon}(f) = S_{x,\epsilon}\}$, i.e., on a set of $\tilde{x}^s$ that are in the sensibly reversed stage. However, in spite of the existence of such a flat loss area, the model still can learn with $\tilde{x}^s$ in sensibly reversed stage. This is obvious because the cross-entropy loss has its non-zero gradient when it is $\log \frac{1}{c}$. Interestingly, when the model is updated in a way to decrease the loss of the previous $\tilde{x}^s$, the new sensible perturbation is again adapted to have the loss approximately equal to $\log \frac{1}{c}$. Therefore, the course of training directs the model to have sensible adversarial examples in the full PGD stage. Furthermore, as long as $c > 0$, even when $\tilde{x}^s$ is in a sensibly reversed stage, it still has perturbation in an adapted $\epsilon$-ball. This helps to obtain a robustness although it is not on a full $\epsilon$ ball.

Note that for $c \geq 0.5$, the learning is very stable because only natural examples can overpower the training; The large loss values are only achievable by sensible attack in the natural stage. Any single full PGD attack cannot dominate the next update of the training because the loss function has relatively small gradients on the full PGD stage. Therefore, our algorithms not only effectively ignore any influential full PGD attacks that may overpower the next update but also train a model that allows as many sensible attacks to be full PGD attacks. In other words, our algorithms can stably learn a robust model.

# E  HIGH MARGIN PROPERTY OF SENSIBLE ADVERSARIAL TRAINING

In sensible adversarial learning, the natural accuracy clearly takes priority over the adversarial accuracy. The perturbed example is not allowed to cross the decision boundary of the Bayes rule in (2) or to reach the vicinity of the decision boundary of a reference function in (4). As mentioned, when the cross-entropy loss is used, $\hat{p}_{f,y}(x) \geq c$ is a prerequisite for adding any adversarial perturbation on $x$. We note that this condition provides

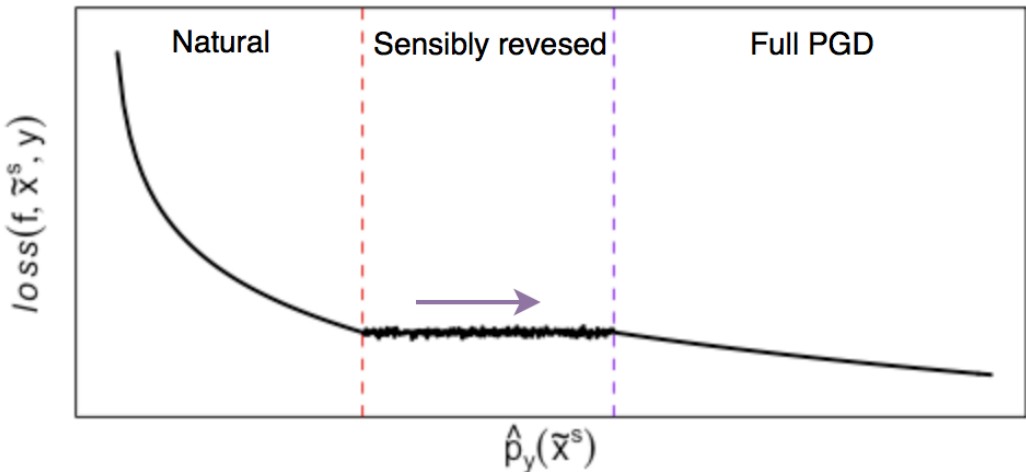

Figure 4: An abstraction of the sensible adversarial loss when $c \geq 0.5$. When $x$ is in the sensibly reversed stage for a current model $f$, the loss of $\tilde{x}^s$ is approximately $\log \frac{1}{c}$. Although the loss is approximately the same while $x$ stays in this stage, the model updates in a way to pushes the sensible adversarial example to become a full PGD attack as the arrow.

a lower bound of the following natural margin as

$$M(f, x, y) = \log \hat{p}_{f,y}(x) - \max_{y' \neq y} \log \hat{p}_{f,y'}(x) \geq \log \frac{\hat{p}_{f,y}(x)}{1 - \hat{p}_{f,y}(x)} \geq \log \frac{c}{1 - c}.$$

Therefore, the priority of natural accuracy hints that the learning will prevent the natural margin from being sacrificed for the sake of adversarial robustness. We note that the natural margin of $x$ is an upper bound of its adversarial margin. Therefore, if a model cannot confidently predict a natural example $x$, neither can the model confidently predict any adversarial examples of $x$.

**Experiment 1**   We investigate the margins of the models trained in Experiment 2, to understand the effect of $c$. In Figure 5 we draw density plots of the margins on the test set for varying $c$ for the fixed capacities. Overall, we see that a larger $c$ results in a larger adversarial margins and natural margins. In Figure 6, we also draw the margins but for varying model capacity. In general, for a each $c$, a smaller capacity has more data points of negative margins. However, for naturally trained models, i.e., the models with $c = 1.0$, a larger model has smaller adversarial margins. This is consistent to the observation of Su et al. (2018) that accurate models tend to be less robust when the models are trained with natural examples. On the other hand, although not displayed, the plots corresponding to $c = 0.9$ are essentially similar to the plots in the second low. This implies even for large $c$, our method is not like natural learning; The models trained with $c = 0.9$ still have larger natural and adversarial margins.

Note that in Figure 3, the regular PGD model of capacity 5 has only few negative adversarial margins. Instead, their natural margins significantly smaller than those of the natural models. In contrast, our model has negative adversarial margins for a few more data points. Note that the majority of both natural and adversarial margins of the our models are significantly larger than that of the regular PGD models. For capacity 1,2 and 3, the regular PGD models collapse having small "mean" of adversarial losses. On the contrary, our models deal with the lack of model capacity by letting more portion of examples to have negative margins. As demonstrated by Figure 5, in our methods the mean adversarial loss can be arbitrarily large. Instead, it maintains a large portion of points having relatively large adversarial margins, i.e., being far from decision boundaries.

The flip side of the advantage of high margin is the possibility of over fitting. For $c \geq 0.5$ in Figure 7, the best robust accuracies are not achieved when capacity is 5 but when capacity is 3. On the contrary, the PGD method achieves better robustness as the capacity increases. Therefore, there is a possibility of over-fitting problem that arises form the high margin property of sensible adversarial learning.

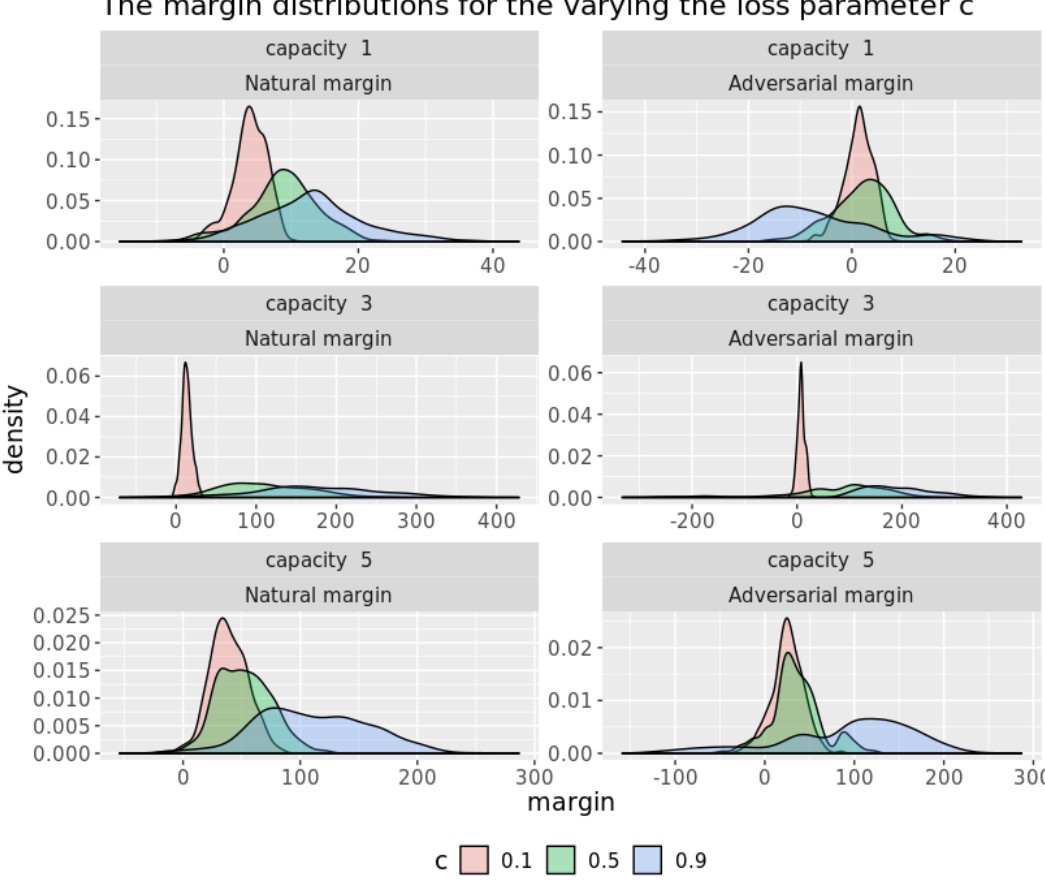

Figure 5: The prediction margins at convergence of the models in Experiment 1 on the test set.

**Experiment 2** In Figure 8, we draw the plot of natural and adversarial margins on the testset of CIFAR10. Compared with the distribution of the TRADE model, the natural margin of the SENSE model is large. For the adversarial margins, the SENSE model has two clearly separated clusters; one is of negative margins and the other is of positive margins. Instead, the positive margins are distributed on the larger values. We remark that this phenomenon is consistent to the sensible idea to allow to be fooled near the decision boundary of the Bayes rule. When the capacity is not enough, the concept of the decision boundary is not of the Bayes rule. Its is projected to an inevitable boundary of a model with nice natural performance, caused by the lack of model capacity. The portion of negative adversarial margin is a cost for sensibility and with this cost, the model can obtain robustness as much as possible given a model capacity. We see the portion of adversarial margins of the SENSE model in the negative area in Figure 8 is not small. This may imply the current model capacity and the sample size for SENSE are not enough.

## F    COMPARISON WITH OTHER METHODS

The sensible selection and reversion in our approach distinguish sensible learning from other approaches that balance between the natural and adversarial accuracy. The objective function of our algorithm can be rewritten as follows. For $f \in \mathcal{F}$,

$$\mathcal{L}(f) = \frac{1}{n} \sum_{i=1}^{n} \ell^s(f, x_i, y_i)$$
$$= \frac{|A_f|}{n} \hat{\mathcal{R}}_{std}(f|A_f) + \frac{|B_f|}{n} \hat{\mathcal{R}}^s_{rob}(f|B_f) + \frac{|C_f|}{n} \hat{\mathcal{R}}_{rob}(f|C_f), \tag{11}$$

where $\ell^s(f, x, y)$ is defined in (5).

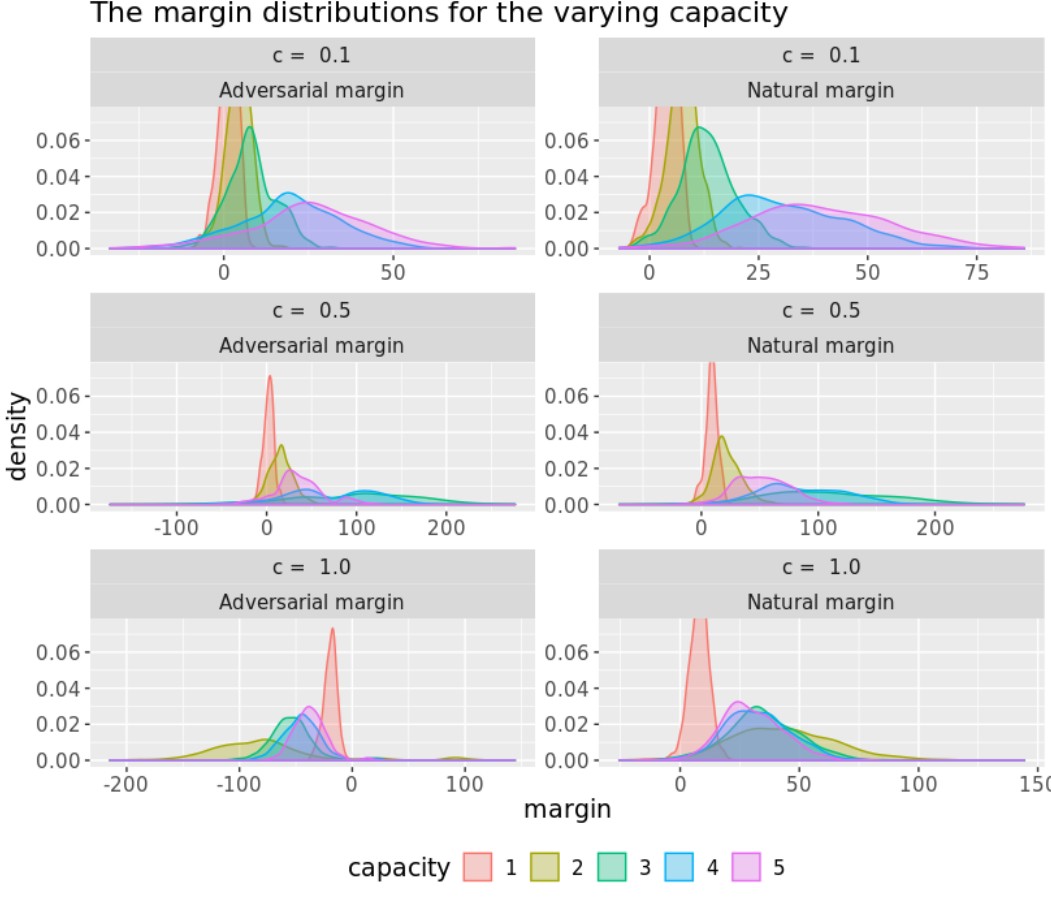

Figure 6: The prediction margins at convergence of the models in Experiment 1 on the test set.

**Kurakin et al. (2016b)** suggest adversarial training that randomly divides data into two parts, a natural and adversarial set. The objective function can be written as the following. For random index sets $A$ and $C$ s.t. $|A| + |C| = n$ and $A \cap C = \phi$,

$$\mathcal{L}(f) = \frac{1}{|A| + \lambda|C|} \Big[ \sum_{i \in A} \ell(f(x_i), y_i) + \lambda \sum_{i \in C} \ell(f(\tilde{x}_i), y_i) \Big]$$

$$= \frac{|A|}{|A| + \lambda|C|} \hat{\mathcal{R}}_{std}(f|A) + \lambda \frac{|C|}{|A| + \lambda|C|} \hat{\mathcal{R}}_{rob}(f|C), \tag{12}$$

where $\tilde{x}_i$ is an adversarial example of $x_i$.

Our approach is similar in that we also divide the data for different usage. However, we divide the data by not a random but an *adaptive* way, so that $\ell^s(f, x_i, y_i) \geq \ell^s(f, x_j, y_j)$ for $x_i \in A_f$ and $x_j \in C_f$. Also, we have an additional set other than a natural and fully adversarial set. This additional set, a sensibly reversed adversarial set, plays an important role in allowing a data point smoothly changes its identity between a natural example and a full adversarial example. Note that we do not have any weight controllers like $\lambda$ in (12). The hyperparameter $c$ itself controls the importance of the natural accuracy in comparison to the adversarial accuracy.

**Zhang et al. (2019)** investigate the Bayes decision boundary to resolve the trade-off problem. They propose TRADE, of which the objective function is for $\beta > 0$,

$$\mathcal{L}(f) = \mathbb{E}\Big[\ell(f(X), y) + \beta \max_{X' \in B(X, \epsilon)} \ell(f(X'), f(X))\Big]. \tag{13}$$

This formulation shows several key differences between TRADE and SENSE. First, (13) uniformly restricts the perturbation norm to $\epsilon$ for all data points, whereas (11) selects the sets for $A_f$, $B_f$ and $C_f$ and restricts

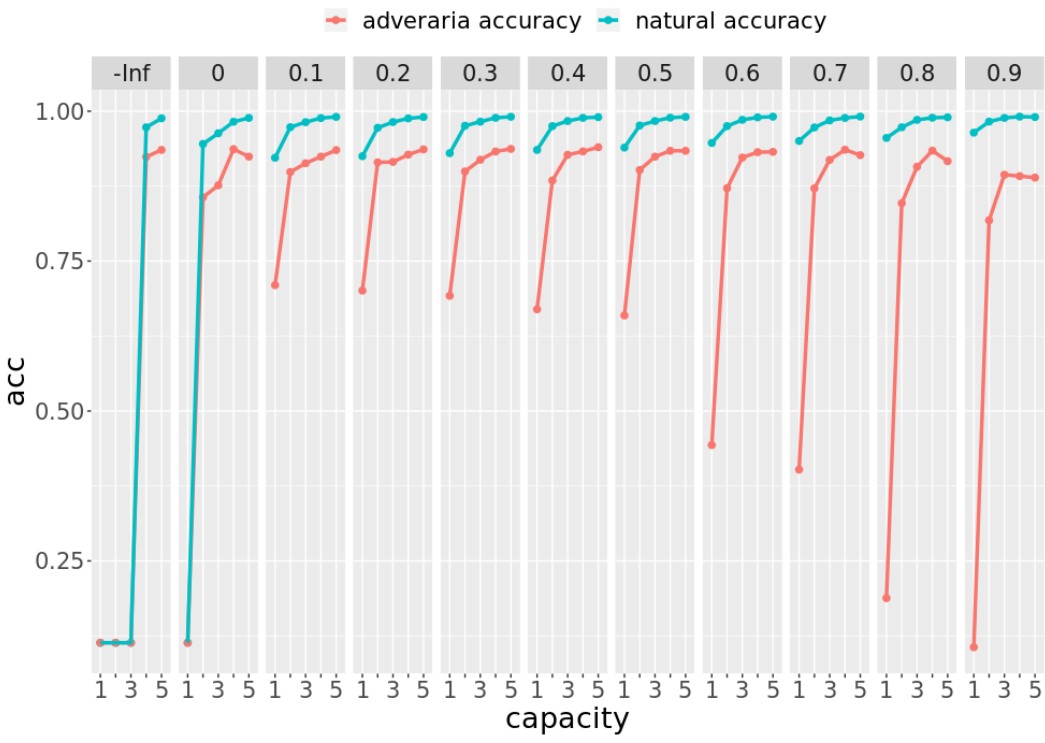

Figure 7: Another visualization of Figure 2. The name of each panel denotes the hyperparameter $c$. -Inf denotes the model trained with the PGD method.

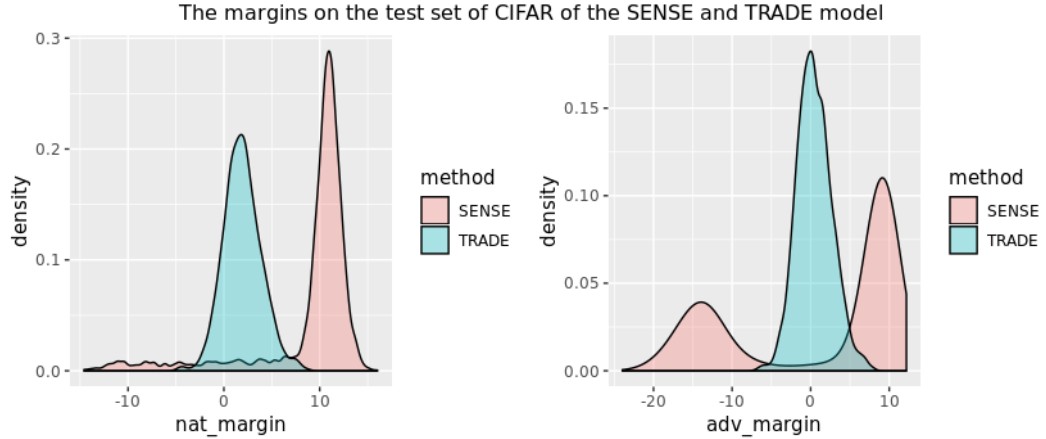

Figure 8: Adversarial and natural prediction margins on CIFAR10 of the SENSE and TRADE model. The margins are calculated by $M(f, x, y) = \log \hat{p}_y(x) - \max_{y' \neq y} \log \hat{p}_{y'}(x) = s_y(x) - \max_{y' \neq y} s_{y'}(x)$, where $s$ denote a score function of $f$, i.e., the output of the neural network.

Table 6: MNIST: test results of the our models on natural examples and $\ell_\infty$ based attacks.

| Defence model | $\epsilon = 0.3$ | $\epsilon = 0.33$ | $\epsilon = 0.36$ | $\epsilon = 0.39$ |
|---|---|---|---|---|
| SENSE | 96.46 | 92.89 | 83.15 | 63.51 |
| TRADE | 96.72 | 90.56 | 46.94 | 11.66 |

Table 7: CIFAR: test results of the our models on natural examples and $\ell_\infty$ based attacks.

| Defence model | $\epsilon = 10/255$ | $\epsilon = 12/255$ | $\epsilon = 14/255$ | $\epsilon = 16/255$ |
|---|---|---|---|---|
| SENSE | 62.63 | 60.39 | 58.05 | 55.09 |
| TRADE | 47.61 | 39.05 | 31.57 | 25.15 |

the perturbation in different ways. Second, the main parameter $\beta$ in (13) controls the importance of the natural accuracy in comparison to the smoothness of the model. However, the main parameter $c$ in (11) directly controls the lower bound of the natural and adversarial loss of the individual data. This is the lower bound on the prediction probability $c$ if $\ell$ is the cross entropy loss. Third, the term $\ell(f(X'), f(X))$ in (13) leads the model to be smooth to all directions in the input space, but $\ell^s(f, x, y)$ in (11) leads to be close to one. Therefore, intuitively, TRADE achieves robustness by obtaining smoothness of the model, whereas SENSE achieves robustness by reformulating (1) in a way to promote high confidency of the robust prediction. This may provide an intuition on the plot of the margin density in Figure 8. By this intuition, we apply PGD attacks with larger perturbation of the training $\epsilon$. We apply the attacks on our trained models and the TRADE model by Zhang et al. (2019) for MNIST and CIFAR10. The results are in Table 6 and Table 7.

## G   ADDITIONAL INFORMATION ABOUT CHEESE HOLE DISTRIBUTION

Figure 9 (e) compares the worst-case standard risk $\mathcal{R}_{std}(\tilde{f}^*_{rob})$ and $\mathcal{R}_{std}(\tilde{f}^{B*})$. Although $\mathcal{R}_{std}(\tilde{f}^*_{rob})$ does not consistently decrease as $\epsilon$ increases, it is always much smaller than $\mathcal{R}_{std}(\tilde{f}^{B*})$. Therefore, pursuit of robustness leads to a more naturally accurate classifier. Figure 9 (e) shows that, as $\epsilon$ increases from $\epsilon = \alpha/2$ to 1, the standard risk of the robustly optimal model gradually increases whereas the sensibly robust model keeps zero risk. Figure 9 (f) demonstrates the robustness against $\epsilon$-ball attacks of $\tilde{f}^{B*}_{nat}$ (black), $\tilde{f}^*_{rob}$ (blue), and $\tilde{f}^{s*}_{rob}$ (red). Although sensibly robust models have large adversarial robustness increasing to 1 as $\epsilon$ increases, this is because more and more adversarial examples can cross the border, while the model keeps its decision boundary consistent to the class border line. On the other hand, adversarially robust functions have constant robust risk for $\epsilon > 1/4$. This is because the robust functions predict as $y = 1$ for every $x \in \mathcal{X}$.

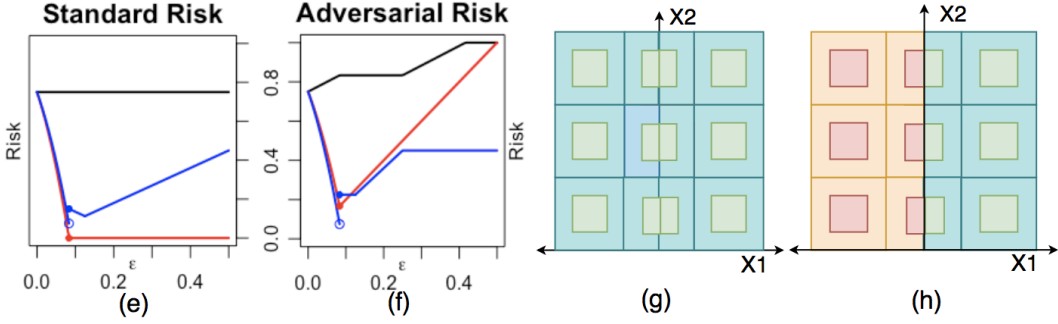

Figure 9: Cheese holes distribution. (e) and (f) The natural and robust risk when $p = 0.55$. The black, blue, and red colors are the worst cases of naturally, adversarially, and sensibly robust functions. (g) The robustly optimal model when $\epsilon > 0.25$. (h) The sensibly robust model when $\epsilon > 1/12$.

**The sketch of the proof on the standard and adversarial robust risks in (e) and (f) in Figure 9**
We first calculate the three classes of functions which minimize natural, adversarial robust, and sensibly robust risk respectively, w.r.t $\tilde{\mathbb{P}}_{X,Y}$. Then for each class, we consider the worst case function from each class, in that the function maximizes the *standard* risk w.r.t $\mathbb{P}_{X,Y}$. The corresponding *standard* risks are in Figure 9 (e).

Likewise, for each class, we consider the worst case function from each class, in that the function maximizes the *adversarial robust* risk w.r.t $\mathbb{P}_{X,Y}$. The corresponding *adversarial robust* risks are in Figure 1 (f).

First, the minimizers of each risk w.r.t. $\tilde{\mathbb{P}}_{X,Y}$ are as following.

1) Let $\tilde{\mathcal{F}}_B$ be a set of naturally optimal functions w.r.t $\tilde{\mathbb{P}}_{X,Y}$:

$$\tilde{\mathcal{F}}_B = \{f \in \mathcal{F} | f(x) = sign(x_1 - 0.5) \text{ for } (x_1, x_2) \in \tilde{\mathcal{X}}\}$$

2) Let $\tilde{\mathcal{F}}_{rob}^s$ be a set of sensibly optimal functions w.r.t $\tilde{\mathbb{P}}_{X,Y}$:

$$\tilde{\mathcal{F}}_{rob}^s = \{f \in \mathcal{F} | f(x) = sign(x_1 - 0.5) \text{ for } (x_1, x_2) \in B(\tilde{\mathcal{X}}, \epsilon)\}.$$

3) Let $\tilde{\mathcal{F}}_{rob}$ be a set of robustly optimal functions w.r.t $\tilde{\mathbb{P}}_{X,Y}$:

$$\tilde{\mathcal{F}}_{rob} = \{f \in \mathcal{F} | f(x) = g(x) \text{ for } (x_1, x_2) \in B(\tilde{\mathcal{X}}, \epsilon)\},$$

where

$$g(x) = \begin{cases} sign(x_1 - 0.5 + \epsilon) \text{ if } \epsilon < \alpha/2 \text{ or } \epsilon \geq 3\alpha/4 \\ 1_{(x_1 \geq 0.5 - \epsilon)} - 1_{(x_1 \leq 3\alpha/2 + \epsilon)} \text{ if } \alpha/2 \leq \epsilon < 3\alpha/4 \end{cases} \tag{14}$$

Second, we consider the worst case standard risk w.r.t. $\mathbb{P}_{X,Y}$ for each class above.

1) $\max_{f \in \tilde{\mathcal{F}}_B} \mathcal{R}_{std}(f)$: Although $p \neq 0.5$, due to the symmetry of the shape of $\tilde{\mathcal{X}}$, $\max_{f \in \tilde{\mathcal{F}}_B} \mathcal{R}_{std}(f)$ is the area on $\mathcal{X} \setminus \tilde{\mathcal{X}}$, the area outside the small nine squares.

2) $\max_{f \in \tilde{\mathcal{F}}_{rob}^s} \mathcal{R}_{std}(f)$: For the same reason above, $\max_{f \in \tilde{\mathcal{F}}_{rob}^s} \mathcal{R}_{std}(f)$ is the area on $\mathcal{X} \setminus B(\tilde{\mathcal{X}}, \epsilon)$, the area outside the small nine squares extended by $\epsilon$.

3) $\max_{f \in \tilde{\mathcal{F}}_{rob}} \mathcal{R}_{std}(f)$: When $\epsilon < \alpha/2$ or $\epsilon \geq 3\alpha/4$, we consider the deviated line on $B(\tilde{\mathcal{X}}, \epsilon)$, and regard the model outside $B(\tilde{\mathcal{X}}, \epsilon)$ as incorrect. The risk is calculated easily by using the fact that the risk of any $f \in \tilde{\mathcal{F}}_{rob}$ on is $B(\tilde{\mathcal{X}}, \epsilon)$ is $3 \times \min(a + 2\epsilon, 2a) \times (1 - p) \times \min(2\epsilon, 0.5)/0.5$. When $\alpha/2 \leq \epsilon < 3\alpha/4$, since $B(\tilde{\mathcal{X}}, \epsilon)$ covers $\tilde{\mathcal{X}}$, the worst case functions are in a form of $f(x) = sign(x_1 - 0.5 + c_\epsilon)$ for some $c$ on the entire $\tilde{\mathcal{X}}$. For each $\epsilon$ s.t. $3\alpha/2 + \epsilon < x_1 < 0.5 - \epsilon$, it is easy to find the corresponding $c_\epsilon$.

Last, for the worst case adversarial robust risk w.r.t. $\mathbb{P}_{X,Y}$, we can calculate the risks in a similar way to above.

# H   ADDITIONAL INFORMATION ABOUT EXPERIMENTS

## H.1   EXPERIMENT 1

We consider the MNIST and CIFAR-10 dataset (LeCun et al., 2010; Krizhevsky & Hinton, 2009).

**Training**   For each dataset, we initialize our model by a naturally trained model. Then, we train the initialized model with sensible adversarial examples with the specifications in Table 8.

Table 8: The learning specifications for the SENSE models in experiment 1

| Dataset | $\epsilon$ | $\eta_1$ | $K$ | $c$ | Initial $\eta_2$ | Epoch |
|---|---|---|---|---|---|---|
| MNIST | 0.3 | 0.05 | 10 | 0.5 | 0.01 | 500 |
| CIfAR10 | $\frac{8}{255}$ | $\frac{8}{255} \times \frac{2}{10}$ | 10 | 0.7 | 0.1 | 300 |

**Testing with white-box attacks**   For white-box attacks, we consider PGD (Madry et al., 2017), C&W (Ding et al., 2019a), DeepFool (Moosavi-Dezfooli et al., 2016), FGSM (Kurakin et al., 2016a), LBFGS (Tabacof & Valle, 2016), and MIFGSM (Dong et al., 2018). In Experiment 1, we consider adversarial perturbations with $\ell_\infty$-norm less than $\epsilon$, where $\epsilon = 0.3$ for the MNIST dataset and $\epsilon = 8/255$ for the CIFAR10 dataset.

We attack our models with the white-box attacks. We use the attacks implemented in Foolbox (Rauber et al., 2017), Advertorch (Ding et al., 2019b), and Adversarial Robustness 360 Toolbox (ART) (Nicolae et al., 2018). The attack specifications are in Table 9. The options that are not listed in the table are kept as default of the attack generating functions.

Table 9: The white-box attack specifications. We denote the step size and step number by $\eta_1$ and $K$.

| Dataset | Attack | $\eta_1$ | $K$ | Python package | Function |
|---|---|---|---|---|---|
| MNIST | PGD500 | 0.01 | 500 | Advertorch | LinfPGDAttack |
| MNIST | C&W40 | 0.01 | 40 | ART | CarliniLInfMethod |
| CIfAR10 | PGD100 | $\frac{2}{255}$ | 100 | Advertorch | LinfPGDAttack |
| CIfAR10 | C&W40 | $\frac{8}{255} \times \frac{1}{20}$ | 40 | ART | CarliniLInfMethod |
| CIfAR10 | DeepFool | default | default | Foolbox | DeepFoolLinfinityAttack |
| CIfAR10 | FGSM | $\frac{8}{255}$ | 1 | Advertorch | GradientSignAttack |
| CIfAR10 | LBFGS | default | default | Foolbox | LBFGSAttack |
| CIfAR10 | MIFGSM | $\frac{8}{255} \times \frac{1}{40.001}$ | 40 | Foolbox | MomentumIterativeAttack (distance=Linfinity, return_early=False) |

We set the step size for MIFGSM as slightly smaller than $\frac{8}{255} \times \frac{1}{40}$ in order to keep the adversarial example from having the perturbation norm greater than $\epsilon = 8/255$. For the performance of TRADE on C&W40, we apply the C&W40 attack with the same specifications in Table 9. For the other results of TRADE in Table 1, we refer to Zhang et al. (2019). For PGD attacks, we draw accuracy and loss plots for increasing step numbers in Figure 10 and Figure 11. They show that for the chosen step sizes, the chosen step numbers are enough to generate proper local maximizer of the PGD objective function. Particularly, for CIFAR, we use the step size $2/255$ rather than $\epsilon \times 2/255$ as it is more efficient.

**PGD attack serenity check**  We test with 100 random restarts, and for the step size 0.01, the step number 500 seem enough by Figure 10.

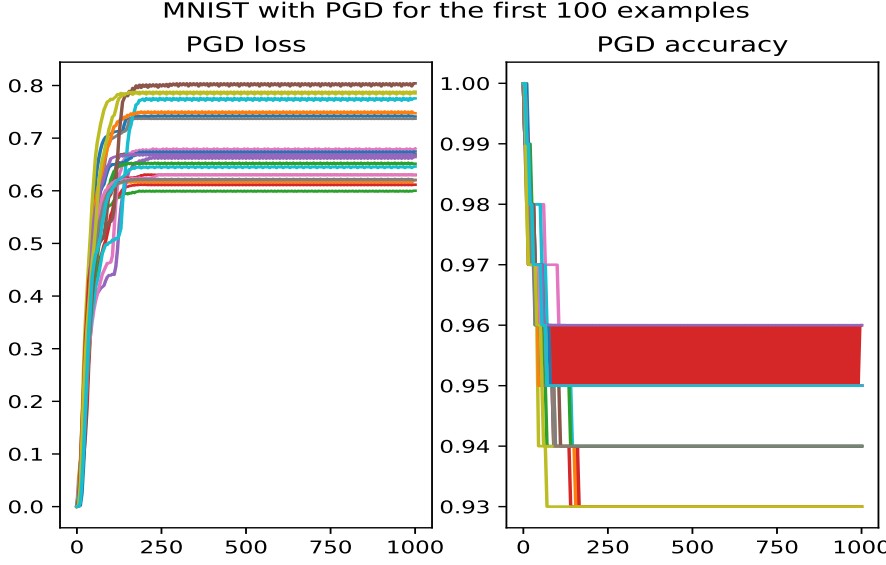

Figure 10: The convergence check for the PGD attacks on the MNIST model. We used a step size 0.01. We can see that $K = 500$ is enough to achieve the lowest point by counting the worst case of random restarts.

**Testing with black-box attacks**  We attack our models with PGD40 and MIFGSM swith the specifications in Table 9. As the Foolbox implementation for MIGSM only returns the successful attacks on the generating model, we only apply these attacks on the defense model. We note that the argument return_early of MIFGSM is set to False as in Table 9. For TRADE, we use the models by Zhang et al. (2019) for both MNIST and CIFAR10.

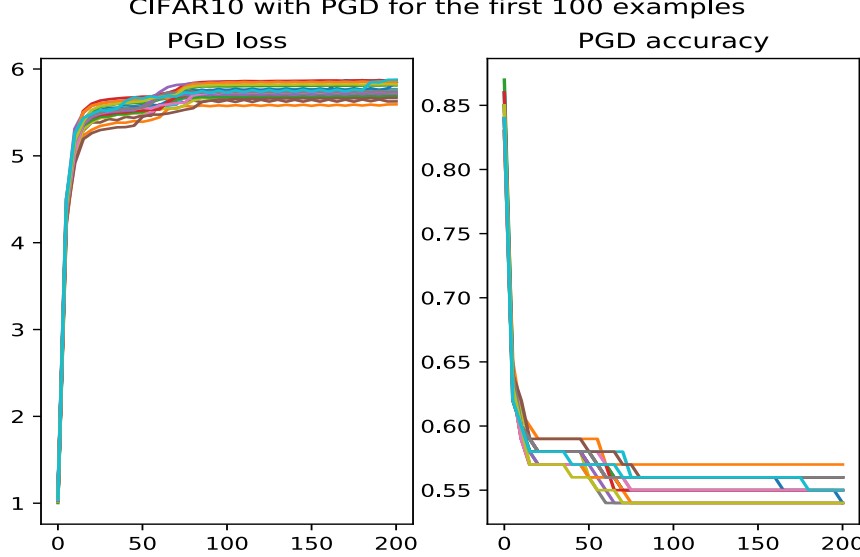

Figure 11: The convergence check for the PGD attacks on the CIFAR model. We used a step size 0.01 and for CIFAR, 2/255. We can see that $K = 100$ is enough to achieve the lowest point by counting the worst case of random restarts.

## H.2    EXPERIMENT 2

**Model Architecture**    We conduct Experiment 2 on the MNIST dataset (LeCun et al., 2010). We consider a sequence of CNNs with the increasing number of kernels. A network of capacity $q$ has two convolutional layers with $2^{(d-1)}$ and $2^d$ filters respectively, followed by a fully connected linear layer of $2^{(d+4)}$ units. Each layer is activated by ReLU. Each convolutional layer is followed by $2 \times 2$ a max-pooling layer. The size of all convolutional filters is $5 \times 5$.

With a similar sequence of CNNs, Madry et al. (2017) investigate the model behavior when the capacity increases. They have capacity scale 1,2,4,8 and 16. In their experiment, capacity scale 1 and 2 collapse. our capacities 2 and 3 are comparable to the capacity scale 1 and 2 by Madry et al. (2017). Likewise, our capacities 4 and 5 are comparable to their capacity scale 4 and 8[1]. Therefore, our result, which shows the PGD models of capacity 1,2 and 3 collapse, is consistent to the result by Madry et al. (2017).

**Training**    We train the sequence of MNIST models with sensibly adversarial example with $\epsilon = 0.3$, $\eta_1 = 0.05$ and $K = 10$ for varying $c \in \{0.0, 0.1, \cdots, 0.9\}$. The initial learning rate $\eta_2$ is 0.01, and we train for 500 epochs. When training the PGD models, we use the same hyperparameters except $c$.

**Testing**    The MNIST models are tested with $\ell_\infty$ PGD attacks of $\epsilon = 0.3$ with the step number $K = 40$ and step size $\eta_1 = 0.01$. We generate the attacks by using a Python package Advertorch by Ding et al. (2019b).

## H.3    ADDITIONAL EXPERIMENT FOR TABLE 6 AND TABLE 7

We conduct our additional experiment on the MNIST and CIFAR-10 dataset (LeCun et al., 2010; Krizhevsky & Hinton, 2009). For each dataset, we consider the SENSE model trained in Experiment 1 and the TRADE model by Zhang et al. (2019). We note that for each dataset, the TRADE and SENSE model share the same architecture. On theses models, we apply the PGD attacks with perturbations larger than the training $\epsilon$. We generate the PGD attacks, using a Python package Advertorch by Ding et al. (2019b) with the following attack specifications.

**MNIST**    Let $\epsilon_0 = 0.3$, which is the training $\epsilon$ for each model. For $\delta \in \{1.1, 1.2, 1.3\}$, we generate the $\ell_\infty$ PGD attacks of $\epsilon = \delta\epsilon_0$ with the step number $K = 40 \times \delta$ and step size $\eta_1 = 0.01$.

---

[1]Our models could be slightly smaller than the counterparts by Madry et al. (2017) because our max-pooling layers do not apply any padding.

**CIFAR-10**    Let $\epsilon_0 = 8/255$, which is the training $\epsilon$ for each model. For $\delta \in \{\frac{10}{8}, \frac{12}{8}, \frac{14}{8}, \frac{16}{8}\}$, we generate the $\ell_\infty$ PGD attacks of $\epsilon = \delta\epsilon_0$ with the step number $K = 40 \times \delta$ step size $\eta_1 = \epsilon_0/20$.

