# OpenReview forum: "Sensible adversarial learning"
_ICLR.cc/2020/Conference — Reject_

### Official Review · AnonReviewer2 · 2019-10-27
**Official Blind Review #2**

**Rating:** 6

**Review:**

Motivated from the so-called trade-off between robustness and standard accuracy in the existing adversarial learning, this paper has proposed a "sensible" adversarial example framework without losing  significantly  performance in natural accuracy. Some toy examples have been presented, showing its reasonableness of the model. The proposed algorithm looks very simple, but it appears that it could be effective through some experiments on two data sets.

Though the reviewer did not fully understand all the details, the proposed idea  seems reasonable. In general, the paper chose those adversarial examples that won't change the predication class. Such adversarial examples, called sensible adversarial examples, mean the perturbation which may not mislead the decision boundary.

There are two concerns with the paper from the reviewer.

(1) The paper states that there is always a trade-off between between robustness and standard accuracy in the adversarial learning work, this seems arguable to the reviewer. In some adversarial example learning literatures, the adversarial learning appears able to improve the performance of both natural examples and adversarial examples.  For example, the VAT approach published in PAMI. The reviewer would like to see more comments about this.

(2) Though there are some theoretical analysis in the paper, the empirical validation may not be very convincing. MNIST and CIFAR 10 are relatively easier datasets, I am not sure if a same observation can be attained when the algorithm is applied on some more complicated and challenging dataset. It remains unclear to me if the "sensible" way could indeed be consistently useful in practical and more challenging cases.


**Experience Assessment:**

I have published one or two papers in this area.

**Review Assessment: Checking Correctness Of Derivations And Theory:**

I did not assess the derivations or theory.

**Review Assessment: Checking Correctness Of Experiments:**

I assessed the sensibility of the experiments.

**Review Assessment: Thoroughness In Paper Reading:**

I read the paper at least twice and used my best judgement in assessing the paper.

---

> ### Author Response · Authors · 2019-11-15
> **Response to Reviewer 2**
>
>
> (1) The paper states that there is always a trade-off between between robustness and standard accuracy in the adversarial learning work, this seems arguable to the reviewer. In some adversarial example learning literatures, the adversarial learning appears able to improve the performance of both natural examples and adversarial examples.  For example, the VAT approach published in PAMI. The reviewer would like to see more comments about this.
>
> $Response$: Thank you for your comment. In the updated paper, some revision is made to remove the confusion. We do not think there must be a trade-off. For example, in Remark 2, we address a sufficient condition that there is no trade-off under the notion of the regular robustness framework.
>
> For the methods that regularize model smoothness such as VAT, we think the occurrence of robustness trade-off would depend on the relative size of $\epsilon$. If $\epsilon$ is small so that the variation in the prediction by the perturbation can be limited by pursuing a smoothness of a model, these methods could improve both natural and adversarial accuracy. However, for an $\epsilon$ larger than this, achieving robustness may require a stronger smooth penalty, which can potentially even harm the natural accuracy. We note that the same objective function of the VAT approach is already used by TRADE with different optimizations. However, TRADE also has shown a trade-off, although its trade-off is less serious than that of PGD. We have demonstrated that SENSE is better than TRADE in terms of trade-off.
>
> (2) Though there are some theoretical analysis in the paper, the empirical validation may not be very convincing. MNIST and CIFAR 10 are relatively easier datasets, I am not sure if the same observation can be attained when the algorithm is applied on some more complicated and challenging dataset. It remains unclear to me if the "sensible" way could indeed be consistently useful in practical and more challenging cases.
>
> $Response$: Thank you for the comments. We have performed more extensive empirical studies to demonstrate the effectiveness of our method.
>
> First, we have tested our trained models more thoroughly with random 20 restarts for PGD attacks and updated the results. We also conducted the same test on the model that is regarded as current state of the art in the trade-off problem.
>
> $$\begin{array}{|c|c|c||c|c|c|}
> \hline
>  \text{MNIST}& \text{SENSE} & \text{TRADE}& \text{CIFAR}& \text{SENSE} & \text{TRADE}\\ \hline
> \text{PGD500 (step=0.01)} &92.21&93.68&\text{PGD100 (step=2/255)}&57.23&54.72\\\hline
> \end{array}$$
>
> Second, we have added the sensitivity analysis on $c$ for the CIFAR model.
>
> $$\begin{array}{|c||c|c|c|c|c|c|}
> \hline
>  \text{CIFAR}& c=0.0 & c=0.3&c=0.5& c=0.6& c=0.7&c=0.8 \\ \hline\hline
> \text{Natural data} &82.88&86.76&90.42&90.87&91.51&\textbf{92.35}\\ \hline
> \text{PGD-100},\eta_1 &43.70&46.90&50.95&55.90&\textbf{57.80}&55.60\\ \hline
> \end{array}$$
>
> Third, we also have applied our approach in training the CIFAR model of much smaller model capacity.
> $$\begin{array}{|c||c|c||c|c||c|c||c|c||c|c|}
> \hline
> c \text{ value} &0.1&0.5&0.9\\\hline  \hline
> \text{natural accuracy(%)}& 66.26&75.7&82.02\\ \hline
> \text{PGD40 accuracy(%)}&26.67&20.26&3.95\\ \hline
> \end{array}$$
> This smaller model is exactly the model that has widely been used for adversarial learning for the MNIST datset. The result is stable in that it keeps high natural accuracy while improving robustness of the standard model. For more complex and challenging datasets, in adversarial learning, one of the biggest difficulty is the increasing requirement of a large model capacity as the data become larger and more complex. Therefore, our stable result implies a possibility of scaling-up the sensible algorithm to more complex dataset.
>
> We agree that the MNIST and CIFAR dataset could be relatively easier datasets. However, in the literature, the trade-off of adversarial learning has been an open problem even for MNIST and CIFAR. We believe our work has contributed to move forward to the next step, that is, resolving trade-off on more complex and challenging dataset. We are currently testing our method on Tiny-ImageNet, which is a much more complex dataset. Compared with CIFAR10, Tiny-ImageNet has larger dimensions $(3\times 64\times 64)$, 20 times more classes (total 200 classes), and higher ranks in the images [1].  We will report the final numbers after we finish the training.
>
>
> [1] Yuzhe Yang, Guo Zhang, Dina Katabi, Zhi Xu, ME-Net: Towards Effective Adversarial Robustness with Matrix Estimation. ICML 2019.

---

### Official Review · AnonReviewer1 · 2019-10-28
**Official Blind Review #1**

**Rating:** 8

**Review:**

The paper proposes the notion of a "sensible" adversary that does not perturb data points on which the Bayes-optimal classifier is incorrect. The authors then provide theory showing that minimizing robust risk against such a sensible adversary yields the Bayes-optimal classifier, which addresses the question about standard vs. robust risk posed in prior work. On the experimental side, the authors then introduce a simple yet effective variation of adversarial training / robust optimization. Instead of maximizing the loss over the perturbation set, the proposed variant stops as soon as the loss exceeds a certain threshold. This can be seen as a variant of gradient clipping that reduces the influence of examples with a very high loss. The authors show that their modification yields an 8 - 9% improvement in robust accuracy on CIFAR-10, which gives state-of-the-art performance.

I find the empirical improvements achieved with their modification of PGD-style adversarial training very interesting and recommend accepting the paper. However, it is not clear to me how well the theory is connected to the empirical findings. Moreover, there are additional experiments the authors can conduct to investigate the performance of their method more thoroughly. Concretely:

- The theory relies on the "reference model" f_r being the Bayes-optimal classifier, while the experiments use the current model as reference model. Especially early in training, the current model performs significantly worse than a Bayes-optimal classifier. Moreover, it is unclear if the proposed training modification is effective if the reference model f_r is a separate classifier. It would be interesting to use a separately trained CNN (standard training without any robustness interventions) as a reference model to see if the training modification still yields improvements.

- If the improvements of the proposed method come from the loss of a few adversarial examples dominating the overall loss in a batch, it would be interesting to measure and plot the loss distribution over examples in a batch with experiments.

- As a baseline, comparing the proposed approach to clipping the loss or gradients of each example would be interesting.

- In the robustness evaluation, have you experimented with randomly-restarted PGD?

- To ensure that PGD works as intended, it would be helpful to see a plot of PGD iteration vs. adversarial accuracy.

- It would be interesting to see accuracy numbers (standard and robust) for models trained with different values for the parameter c. This would also provide information about the sensitivity of this hyperparameter.


Further comments:

- I encourage the authors to release their code and pre-trained models in a format that is easy for other researchers to build on (e.g., PyTorch model checkpoints).

- Page 3, "Note that R_rob(f^B) = P(X_1 [...]" - should this be X instead of X_1?

- Line 9 of Algorithm 1: should the sum go from 1 to m?

- Equation 5: is "x <= log 1/c" in the subscript a typo?

- Page 8: "Our model achieves 91.51natural accuracy.": percent symbol and space missing

**Experience Assessment:**

I have published in this field for several years.

**Review Assessment: Checking Correctness Of Derivations And Theory:**

I assessed the sensibility of the derivations and theory.

**Review Assessment: Checking Correctness Of Experiments:**

I assessed the sensibility of the experiments.

**Review Assessment: Thoroughness In Paper Reading:**

I read the paper at least twice and used my best judgement in assessing the paper.

---

> ### Author Response · Authors · 2019-11-15
> **Response to Reviewer 1 (Part 1)**
>
>
> - The theory relies on the "reference model" $f_r$ being the Bayes-optimal classifier, while the experiments use the current model as reference model. Especially early in training, the current model performs significantly worse than a Bayes-optimal classifier. Moreover, it is unclear if the proposed training modification is effective if the reference model $f_r$ is a separate classifier. It would be interesting to use a separately trained CNN (standard training without any robustness interventions) as a reference model to see if the training modification still yields improvements.
>
> $ Response$: Thank you for the suggestion. If we fix a separate trained model as the reference model $f_r$, the robustness improvement would be insignificant compared with that of the reference model. The reasons are summarized as follows:
>
> 1. Given data generated by $\tilde{P}_{X,y}$ with the restricted support $\tilde{\mathcal{X}}$, natural training approximates $f^B$ on $\tilde{\mathcal{X}}$. If this approximation is given as a fixed reference model, the algorithm would end up with a trained model that only approximates $f^B$ on $\tilde{\mathcal{X}}$. On the other hand, if the reference model is updated iteratively, we can achieve robustness on $B(\tilde{\mathcal{X}},\epsilon)\setminus\tilde{\mathcal{X}}$ as well as approximating $f^B$ on $\tilde{\mathcal{X}}$.
> 2. Our algorithmic goal is achievable even when we initialize the model by a naturally trained model that could be significantly worse on $\tilde{\mathcal{X}}^c$ than $f^B$. This is because of our unique loss function that allows gradients even for the reversed adversarial examples. As explained in Appendix D with Figure 4, a reversed adversarial example has the loss value nearly $-\log{c}$, while holding non-zero gradient. This allows the example to have an influence (not excessive) on the next update on the previous decision boundary. While obtaining robustness, the algorithm tries to preserve the accuracy on $\tilde{\mathcal{X}}$ by adding an adversarial perturbation only on a correctly classified point. Therefore, even with an initially non-robust (reference) model, our algorithm can obtain robustness while keeping natural accuracy.
>
>
> - If the improvements of the proposed method come from the loss of a few adversarial examples dominating the overall loss in a batch, it would be interesting to measure and plot the loss distribution over examples in a batch with experiments.
>
> $ Response$: Thank you for the suggestion. In the following link, we have plotted the PGD loss and accuracies of the CIFAR model: https://drive.google.com/file/d/1xeNffZN2vbREkDaWBOv_Kdo0ZtdMFO44/view?usp=sharing.
>
> We note that if correctly optimized, the sensible loss is obtained by trimming the PGD loss. As expected, the loss value distribution of SENSE model has a long right tail, of which the points are potentially most influential points if they were in PGD learning.

---

> > ### Author Response · Authors · 2019-11-15
> > **Response to Reviewer 1 (Part 2)**
> >
> > - As a baseline, comparing the proposed approach to clipping the loss or gradients of each example would be interesting.
> >
> > $ Response$: Thank you for the suggestion. We have tried the followings:
> >
> > 1) Loss clipping: We have tried the loss clipping by minimizing the following loss
> >  $$ \mathcal{L}(\theta)=\sum_{i=1}^n\max(\ell(f_\theta,\tilde{x}_i^{PGD},y_i),C).$$
> >
> > For both MNIST and CIFAR data, we use CNNs that was the architecture used for Table 3 of MNIST.  We initialized the model with a naturally trained model because with random initialization the model did not learn anything, and trained with the same learning rate and epoch to our model.
> >
> > MNIST: We chose the cutting criteria $C=0.7$. We attacked the model with the same PGD 40 attack with step size 0.01 with $\epsilon=0.3$. The training achieved comparable results to the SENSE, TRADE and PGD trained models. The results are as following:
> >
> > natural accuracy : 99\%. adversarial accuracy: 96.8\%.
> >
> > Compared with the SENSE, TRADE and PGD models, the adversarial accuracy is slightly higher and the natural accuracy is slightly lower.
> >
> > CIFAR: Due to the time constraint we chose the same CNNs to that of the MNIST model, which could be not large enough for regular PGD training. As before, we train with $\epsilon=8/255$ with step size $8/255*2/10$ with step number 10. The initial model is naturally trained on the entire training set, which has 80.85\% natural accuracy and 0.0\% adversarial accuracy. We chose $C=-\log(c)$ for $c\in\{0.1,0.3, 0.5, 0.7,0.9\}$. When training with the entire training set, although initialized with model with natural accuracy greater than 80\%, the models of the clipping training collapsed for every $c$ value. Larger the $c$ is slower the collapsing timing was but all within 50 epochs. To reduce the burden of the relatively  complex nature of the CIFAR data, we also tried to train only with 10\% of the training set. However, the same thing happened with more epochs than the entire set. On the contrary, all the sense models did not collapsed. Because all models of the clipping method collapsed with the training set, we only try 10\% training data. We report the results against PGD40 with step size 2/255 here at 300 epochs. Also, we provide the logs and the training and testing code for this additional experiment to in the directory of the initially provided code link. Here, $C$ means collapsed.
> >
> > $$\begin{array}{|c||c|c||c|c||c|c||c|c||c|c|}
> > \hline
> > c \text{ value} &0.1&0.1&0.3&0.3&0.5&0.5&0.7&0.7&0.9&0.9\\\hline
> >  \text{model}&  \text{clip}&\text{sense} & \text{clip}&\text{sense} & \text{clip}&\text{sense} & \text{clip}&\text{sense} & \text{clip}&\text{sense} \\ \hline
> > \text{natural accuracy(%)}& C&65.14&C&68.87&C&71.08&C&74.12&38.6&74.84\\ \hline
> > \text{PGD40 accuracy(%)}&C&15.59&C&10.98&C&11.26&C&7.8&0.0&3.9\\ \hline
> > \end{array}$$
> > Just in case, we add the results of sensible adversarial learning on the entire dataset with $c=0.1, 0.5, 0.9$.
> > $$\begin{array}{|c||c|c||c|c||c|c||c|c||c|c|}
> > \hline  c \text{ value} &0.1&0.5&0.9\\\hline  \hline
> > \text{natural accuracy(%)}& 66.26&75.7&82.02\\ \hline
> > \text{PGD40 accuracy(%)}&26.67&20.26&3.95\\ \hline
> > \end{array}$$
> >
> > $Intuition$: As this loss clipping is similar to the sensible algorithm in that the loss cannot exceed a threshold. The differences is that in sensible learning, any regularized perturbations still has non-zero gradient, while keeping the loss close to the threshold. This is a stark difference from loss clipping, which always gives a zero gradient to the clipped example. Another major difference is that in the sense algorithm, an example in a natural stage can have an arbitrarily large loss value. This keeps the natural accuracy while learning a robust model, preventing collapsing. On the other hand, during the learning of loss clipping. Once a point is ignored by clipping, by fitting other remaining points more, the learning seems to abandon more and more points. Also, it could happen that every loss value is greater than $c$, leading to no gradients for updating in loss clipping. We think this was the reason of collapsing and when a larger capacity is used, the result would be different.
> >
> > 2) Gradient clipping: For gradient clipping, it was not easy to figure out an efficient implementation, and we think it could be an interesting future research direction. A natural way to apply the gradient clipping to adversarial learning would be an example-wise application of gradient clipping. In this case, the clipped gradient could be expressed as $$\sum_{i=1}^n Clip(\partial \ell(f_\theta,\tilde{x}_i^{PGD},y_i)/\partial\theta)$$
> > This implies that the gradient for each example should be clipped before being summed up. However, the implementation seems not obvious because in the current deep learning platforms the derivative is calculated based on the sum of loss values. Manually handling the gradient for each example could be computationally expensive as the batch size increases.

---

> > > ### Author Response · Authors · 2019-11-15
> > > **Response to Reviewer 1 (Part 3)**
> > >
> > > - In the robustness evaluation, have you experimented with randomly-restarted PGD?
> > >
> > > $ Response$: Thank you for the comment. The originally reported PGD-test results were based on without random restarts. After submission we have realized the importance of random restarts for a thorough test with stronger PGD attacks. We report the results with random restarts with all the same attack hyperparameters as the same as before. In the following table, ``no random" columns are the originally reported robust accuracies.
> > >
> > > $$\begin{array}{|c|c|c||c|c|c|c|}
> > > \hline
> > >  \text{MNIST}& \text{no random} & \text{20 randeom restarts}&\text{CIFAR}& \text{no random} & \text{20 random restarts} \\ \hline
> > > \text{PGD-40} &96.46&94.63 & \text{PGD-20}&65.17&62.57\\ \hline
> > > \end{array}$$
> > >
> > > We conduct an additional test that further decreases the robust accuracy of our CIFAR model. As this additional analysis is more relevant to the next comment, we report the accuracy numbers by responding the next comment.
> > >
> > > - To ensure that PGD works as intended, it would be helpful to see a plot of PGD iteration vs. adversarial accuracy.
> > >
> > > $ Response$: Thank you for the suggestion. We present the result in the links and the same plots have been presented in the updated pdf file. We consider the first 100 test examples. In each plot, different line means different random restarts. By plotting as suggested with 20 random restarts, we notice the step numbers chosen for our initial submission are not enough.
> > >
> > > For MNIST (https://drive.google.com/file/d/1r1gRxp3DF1OtOhlDyXnSKhhmTh0udBGw/view?usp=sharing), we plot the loss and accuracies for every other five steps from 0 to 1000. We used the step size 0.01 that was used for attacks in our experiment. We can see that the step number K=40 in our experiment was not enough to generate PGD attacks properly. Considering PGD with random restarts counts the worst case accuracy, we think at least 60 to 250 iterations are necessary for step size 0.01 based on the plots.
> > >
> > > For CIFAR (https://drive.google.com/file/d/19-uM-heJyLDq3_mT-fR7oESZaJoCfQx7/view?usp=sharing), the first 100 examples are used, with two different step sizes. One is $\eta_0=2\epsilon/K$ which was used in our experiment, and the other is $\eta_1=2/255$ that was suggested by Reviewer 3. In the PGD iteration vs. adversarial accuracy plot, we notice that the step size $\eta_1$ is more efficient then $\eta_0$; For $\eta_0$, it is possible not to converge even at 100 iterations. Meanwhile, for $\eta_1$ looks mostly converged before iteration 100, while achieves the same lowest point. Given PGD with random restarts counts the worst case accuracy, we think at least 60 iterations are necessary for step size $\eta_1$ based on the plots.
> > >
> > >   Based on this observation, with 20 random restarts, we tested our models with PGD 500 attacks on our MNIST model and PGD 100 attacks on the CIFAR model. As a baseline, we also conducted the same test on the TRADE model.
> > >
> > > $$\begin{array}{|c|c|c||c|c|c|}
> > > \hline
> > >  \text{MNIST}& \text{SENSE} & \text{TRADE}& \text{CIFAR}& \text{SENSE} & \text{TRADE}\\ \hline
> > > \text{PGD500 (step=0.01)} &92.21&93.68&\text{PGD100 (step=2/255)}&57.23&54.72\\\hline
> > > \end{array}$$
> > >
> > > We have updated Table 1 and 3 of the manuscript accordingly.

---

> > > > ### Author Response · Authors · 2019-11-15
> > > > **Response to Reviewer 1 (Part 4)**
> > > >
> > > >
> > > > - It would be interesting to see accuracy numbers (standard and robust) for models trained with different values for the parameter c. This would also provide information about the sensitivity of this hyperparameter.
> > > >
> > > > $ Response$: Thank you for the comments. While preparing our initial submission, we have trained the CIFAR models with several different $c$ values, where all other hyper parameters were set to the same. By the submitted code, the models can be trained only by changing $c$, except that for $c=0.8$ we stopped learning at 120 epoch whereas others all have 300 epochs. We report the adversarial accuracy against PGD100 attacks. In order to conduct the every test with random 20 restarts in the limited time, we report the results for the first 2000 test examples. We use a step size $\eta_1=2/255$. The step number 100 and step size 2/255 are justified by the plot we draw as an answer of the comment above.
> > > >
> > > > $$\begin{array}{|c||c|c|c|c|c|c|}
> > > > \hline
> > > >  \text{CIFAR}& c=0.0 & c=0.3&c=0.5& c=0.6& c=0.7&c=0.8 \\ \hline\hline
> > > > \text{Natural data} &82.88&86.76&90.42&90.87&91.51&\textbf{92.35}\\ \hline
> > > > \text{PGD-100},\eta_1 &43.70&46.90&50.95&55.90&\textbf{57.80}&55.60\\ \hline
> > > > \end{array}$$
> > > >
> > > > Interestingly, while the natural accuracy positively correlated to the $c$ value, the robustness does not show clear negative correlation to $c$. Rather, as $c$ become closer to 0.7, more robust result the model shows. As we see that $c=0.8$ has better robustness than $c\leq 0.5$, in CIFAR, the main reason of the observed trade-off could attribute to the influential adversarial perturbations.
> > > >
> > > >
> > > > Further comments:
> > > >
> > > > - I encourage the authors to release their code and pre-trained models in a format that is easy for other researchers to build on (e.g., PyTorch model checkpoints).
> > > >
> > > > $ Response$: Thank your for the encouragement. We agree to release our experimental work. We will make the code and pre-trained PyTorch models available on GitHub so that others can easily test our trained models as well as train new models. We note that the code link on our submission page has been available in public and the link contains most of our experimental work including the trained models.
> > > >
> > > > - Typo errors:
> > > >
> > > > $ Response$: Thank you for checking the typos. Yes, you are right in all your comments on the typo errors. To make it clear, in (5) $x\leq \log1/c$ should be $\ell(f,x,y)\leq \log1/c$. We have corrected the typos.

---

### Official Review · AnonReviewer4 · 2019-11-07
**Official Blind Review #3**

**Rating:** 3

**Review:**

Summary:
The paper studies the phenomenon of trade-off between robust and standard accuracies that is usually observed in adversarial training. Many existing studies try to understand this trade-off and show that it is unavoidable. In contrast, this work shows that under a sensible definition of adversarial risk, there is no trade-off between standard accuracy and sensible adversarial accuracy. It is shown that Bayes optimal classifier has optimal standard and sensible adversarial accuracies. The authors then go on to propose a new adversarial training algorithm which tries to minimize the sensible adversarial risk. Experimental results show that models learned through the proposed technique have high adversarial and standard accuracies.

Comments:
1) Sensible Adversarial Risk: The first contribution of the work is to define a new notion of adversarial risk which the authors call ''sensible adversarial risk'' and study its properties.  There is a recent work[1] which also proposes a new definition of adversarial risk and which is similar in spirit to what the current work tries to achieve.  In [1], the authors define a perturbation as adversarial only if it doesn't change the label of the Bayes optimal classifier, which is similar to what the current paper does. Owing to this similarity, the properties of sensible adversarial risk obtained in Theorems 1,2 in the current work look similar to [1]. So the authors should discuss/compare their results with [1].

2) Sensible Adversarial Training: I believe the major contribution of the paper is to propose an algorithm for minimizing sensible adversarial risk. However, I have some concerns with the proposed algorithm. The authors say that since the Bayes optimal classifier is unknown, they use a reference model f_r (which can be naturally trained). Consider the following scenario. Suppose the true data is given by (x, f^B(x)), for some unknown f^B; that is, the Bayes optimal classifier has perfect standard accuracy. Suppose f_r has perfect standard accuracy, but very bad adversarial accuracy on train and test sets. Suppose f_r is substituted for f^B in the sensible adversarial risk (with 0-1 loss). Then it is easy to see that f_r is a minimizer of the resulting objective. So the proposed algorithm will just output f_r in this scenario. This is clearly not desirable. Given this, I believe a more thorough understanding of the proposed algorithm is needed. When will the algorithm converge to non-robust classifiers? How should one initialize the algorithm to avoid such undesirable behavior?

While the notion of sensible adversarial risk is sensible, it is not clear why it should result in such high adversarial accuracies as reported in Tables 1,5. The adversarial perturbation of epsilon=8/255 on cifar10 is considered so small that sensible adversarial risk (with reference model f^B) at any point will almost always be equal to the existing notion of adversarial risk at that point. So, minimizing sensible adversarial risk (assuming you are given f^B) is exactly equivalent to minimizing adversarial risk. But it is known that minimizing adversarial risk results in models with low standard accuracies. So I feel sensible adversarial risk is not the reason behind such high accuracies. Could the authors explain what is the reason for such good adversarial accuracies reported in the paper?

3) Experiments:  While the experimental results on cifar10 look impressive, I have some concerns about the way the PGD attacks are run. I downloaded the model provided by the authors and ran PGD attack on it.  I ran L_infty attacks with epsilon=8/255, step size=2/255. The results I obtained seem to differ from the results presented in the paper:
PGD Steps | Adversarial accuracy of SENSE
20            62.09
50            60.34
100           59.99

I believe PGD attack with multiple random restarts will reduce the adversarial accuracy even further. Given this, I'd appreciate if the authors perform more careful attacks (with appropriate hyper-parameters) on their model. It'd also be great if the authors report the performance of PGD trained model using the same attacks used to report the performance on their model.

4) Other comments:  I'm not sure if the toy example (cheese hole distribution) in Section 2 is helpful. What the authors seem to conclude from it is that adversarial training can improve standard accuracy. But I do not agree with these conclusions. What if Figure 1b is the true distribution? Will the same conclusions hold? In general, these toy examples need not be illustrative of the behavior on real datasets. So instead of having these toy examples, I'd suggest the authors have a thorough discussion on theoretical and experimental results.

[1] Suggala, A. S., Prasad, A., Nagarajan, V., & Ravikumar, P. (2018). Revisiting Adversarial Risk. arXiv preprint arXiv:1806.02924.

**Experience Assessment:**

I have published one or two papers in this area.

**Review Assessment: Checking Correctness Of Derivations And Theory:**

I assessed the sensibility of the derivations and theory.

**Review Assessment: Checking Correctness Of Experiments:**

I carefully checked the experiments.

**Review Assessment: Thoroughness In Paper Reading:**

I read the paper at least twice and used my best judgement in assessing the paper.

---

> ### Author Response · Authors · 2019-11-15
> **Response to Reviewer 3 (Part 1)**
>
>
> 1) Sensible Adversarial Risk: The first contribution of the work is to define a new notion of adversarial risk which the authors call ''sensible adversarial risk'' and study its properties.  There is a recent work[1] which also proposes a new definition of adversarial risk and which is similar in spirit to what the current work tries to achieve.  In [1], the authors define a perturbation as adversarial only if it doesn't change the label of the Bayes optimal classifier, which is similar to what the current paper does. Owing to this similarity, the properties of sensible adversarial risk obtained in Theorems 1,2 in the current work look similar to [1]. So the authors should discuss/compare their results with [1].
>
> $ Response$: Thanks for the reference Suggala et al. and we have added this citation and made an extensive comparison in Section 3.
>
> We agree that we are not the first who tried to formalize the idea of non-class-change condition by using the Bayes rule. However, our way to define the true class change is quite different from that of Suggala et al, and our theorems are orthogonal to that of Suggala et al.
>
> First, the definition of adversarial example is different. Here, for conciseness, we omit the condition on $\delta_x$ s.t. $\|\delta_x\|\leq \epsilon$. Given $\mathcal{Y}\in\{-1,1\}$, for an $f\in \mathcal{R}^\mathcal{X}$ they define an adversarial perturbation as
> $$\delta_x\in argmax_{g(x)=g(x+\delta)}\big[\ell(f(x+\delta_x), g(x))-\ell(f(x), g(x)) \big]    .............     (1)$$
>  Unlike our definition of sensible adversarial examples, their definition on $\delta$ does not involve $y$. If $g$ is a Bayes rule $f^B$, the adversarial goal is to increase the loss w.r.t. not $y$ but a deterministic function of $x$.
>
> Second, the risk is different, and this fact clearly shows the orthogonality between our and their theoretical work.
> Their adversarial risk is
>  $$R_{adv}(f)=\mathbb{E}[\max_{g(x)=g(x+\delta)}\ell(f(x+\delta_x), g(x))-\ell(f(x), g(x))].$$ Their theorem shows that if $g=f^B$, then $f^B$ is the minimizer of $W_\lambda(f)=R_{nat}(f)+\lambda R_{adv}(f).$
>
>  We note that in their theorem 1, $\lambda$ should be strictly smaller than $\infty.$ Otherwise, their Theorem 1 is not established; for $\lambda=\infty$ the problem is equal to minimizing $R_{adv}(f)$. $R_{adv}(f)$ can be minimized by $f$ either if
> 1. $f(x)\neq g(x)$ w.p. 1
> 2. $f(x)\neq g(x)$ or $f(x)= g(x)$, w.p. 1.
>  Therefore, in their theorem, the added $R_{nat}(f)$ term is necessary and $\lambda R_{adv}(f)$ cannot stand by itself. On the contrary, our adversarial risk is defined by using $y$ in the loss function $\ell$. Furthermore, our theorems are established by using the adversarial risk alone. Therefore, the theoretical works of the two papers are orthogonal each other.
>
>  Third, our definition of sensible adversary enables the derivation of our algorithm straightforward. They does not explicitly define $\delta_x$ when $f(x)\neq g(x).$ It could be not-defined, based on their Definition 1, or arbitrary based on above (1). On the other hand, our adversarial examples are well defined on every $x\in\mathcal{X}$, which enables to optimize by data augmentation approaches.
>
> For further information, we refer to our discussions with a public reviewer Elan Rosenfeld below.

---

> > ### Author Response · Authors · 2019-11-15
> > **Response to Reviewer 3 (Part 2)**
> >
> >
> > 2) Sensible Adversarial Training: I believe the major contribution of the paper is to propose an algorithm for minimizing sensible adversarial risk. However, I have some concerns with the proposed algorithm. The authors say that since the Bayes optimal classifier is unknown, they use a reference model $f_r$ (which can be naturally trained). Consider the following scenario. Suppose the true data is given by (x,$f^B$(x)), for some unknown $f^B$; that is, the Bayes optimal classifier has perfect standard accuracy. Suppose $f_r $ has perfect standard accuracy, but very bad adversarial accuracy on train and test sets. Suppose $f_r$ is substituted for $f^B$ in the sensible adversarial risk (with 0-1 loss). Then it is easy to see that $f_r$ is a minimizer of the resulting objective. So the proposed algorithm will just output $f_r$ in this scenario. This is clearly not desirable. Given this, I believe a more thorough understanding of the proposed algorithm is needed. When will the algorithm converge to non-robust classifiers? How should one initialize the algorithm to avoid such undesirable behavior?
> >
> > While the notion of sensible adversarial risk is sensible, it is not clear why it should result in such high adversarial accuracies as reported in Tables 1,5. The adversarial perturbation of epsilon=8/255 on CIFAR10 is considered so small that sensible adversarial risk (with reference model $f^B$) at any point will almost always be equal to the existing notion of adversarial risk at that point. So, minimizing sensible adversarial risk (assuming you are given $f^B$) is exactly equivalent to minimizing adversarial risk. But it is known that minimizing adversarial risk results in models with low standard accuracies. So I feel sensible adversarial risk is not the reason behind such high accuracies. Could the authors explain what is the reason for such good adversarial accuracies reported in the paper?
> >
> > $ Response$:
> > Thank you for your comments. We agree that under 0-1 loss, if the reference model is fixed as the described model $f_r$, then the proposed algorithm will just output $f_r$ in this scenario. This situation has been described in Theorem 4 in Appendix. For the algorithms, we have provided two modifications. First, our algorithm uses the cross entropy loss. Second, our algorithm does not fix the reference function, but instead, updates it iteratively. Specifically,
> >
> > 1. Given data generated by $\tilde{P}_{X,y}$ on the restricted support $\tilde{\mathcal{X}}$, natural training approximates $f^B$ on $\tilde{\mathcal{X}}$. If this approximation is given as a fixed reference model, the algorithm would end up with a trained model that only approximates $f^B$ on $\tilde{\mathcal{X}}$. If the reference model is updated iteratively, we can achieve robustness on $B(\tilde{\mathcal{X}},\epsilon)\setminus\tilde{\mathcal{X}}$ as well as approximating $f^B$ on $\tilde{\mathcal{X}}$.
> > 2. Our algorithmic goal is achievable even when we initialize the model by a naturally trained model that could be significantly worse on $\tilde{\mathcal{X}}^c$ than $f^B$. This is because of our unique loss function that allows gradients even for the reversed adversarial examples.
> >
> > We agree that for a very small $\epsilon$, the sensible robustness is basically same to the adversarial robustness. Our improvement in natural and robust accuracy in this case comes from our algorithmic property. As explained in Appendix D, our sensible reversion effectively regularizes the loss value nearly $-\log{c}$, preventing any influential adversarial perturbation. This property is particularly useful when the model class defined by the model capacity is not enough to have the Bayes rule in it. In addition, given some points that are missclassified or originally near the decision boundary, this property is also useful to well reflect the majority of adversarial and natural distribution, which is helpful for improving both natural and adversarial accuracy.
> > Empirically, we demonstrate the effect regularizing dominating adversarial examples by testing several CIFAR models trained with different $c$, where $\eta_1=2/255$.
> >
> > $$\begin{array}{|c||c|c|c|c|c|c|}
> > \hline
> >  \text{CIFAR}& c=0.0 & c=0.3&c=0.5& c=0.6& c=0.7&c=0.8 \\ \hline\hline
> > \text{Natural data} &82.88&86.76&90.42&90.87&91.51&\textbf{92.35}\\ \hline
> > \text{PGD-100},\eta_1 &43.70&46.90&50.95&55.90&\textbf{57.80}&55.60\\ \hline
> > \end{array}$$
> >
> > The model with better robust accuracy is not the model having the worst natural accuracy. Rather, the models having better natural accuracy hold better robust accuracies. Although it could be strange that natural accuracy increases while robust risk decreases, we think this is because the sensible approach does not aim to directly minimize adversarial risk.

---

> > > ### Author Response · Authors · 2019-11-15
> > > **Response to Reviewer 3 (Part 3)**
> > >
> > >
> > > 3) Experiments:  While the experimental results on CIFAR10 look impressive, I have some concerns about the way the PGD attacks are run. I downloaded the model provided by the authors and ran PGD attack on it.  I ran $L_\infty$ attacks with epsilon=8/255, step size=2/255. The results I obtained seem to differ from the results presented in the paper:
> > > PGD Steps | Adversarial accuracy of SENSE
> > >
> > > 20            62.09
> > >
> > > 50            60.34
> > >
> > > 100           59.99
> > >
> > > I believe PGD attack with multiple random restarts will reduce the adversarial accuracy even further. Given this, I'd appreciate if the authors perform more careful attacks (with appropriate hyper-parameters) on their model. It'd also be great if the authors report the performance of PGD trained model using the same attacks used to report the performance on their model.
> > >
> > > $ Response$: Thank you for bringing our attention to a different step size. We initially chose the step size $2\epsilon/K$ since it seemed almost standard in the literature. However, we found that the suggested step size 2/255 is much more efficient than $2\epsilon/K$ by plotting accuracies vs step numbers for the CIFAR model (https://drive.google.com/file/d/19-uM-heJyLDq3_mT-fR7oESZaJoCfQx7/view?usp=sharing) by the suggestion of Reviewer 1. The plots are the loss and accuracies against PGD attacks on the first 100 test examples, where x-axis is the step numbers and each line denotes different random restart. We observe that the suggested step size 2/255 converges much faster than our previous choice. However, the lower bounds are similar to each other in the end if we consider that the robustness against PGD attacks with random restarts counts the worst case. For our previous step size, even the step number 100 was not enough, but for 2/255, the step number 100 is enough if used with random restarts. Therefore, for the suggested step size, we have tried PGD100 attack with random 20 starts. We have updated Table 1 accordingly and report the results as below:
> > >
> > > $$\begin{array}{|c|c|c|}
> > > \hline
> > >  \text{CIFAR}& \text{SENSE} & \text{TRADE}\\ \hline\text{PGD100 (step=2/255)}&57.23&54.72\\\hline
> > > \end{array}$$
> > >
> > > As a base line, we used the TRADE model as it was known that the TRADE model outperforms the PGD trained model.
> > >
> > > 4) Other comments:  I'm not sure if the toy example (cheese hole distribution) in Section 2 is helpful. What the authors seem to conclude from it is that adversarial training can improve standard accuracy. But I do not agree with these conclusions. What if Figure 1b is the true distribution? Will the same conclusions hold? In general, these toy examples need not be illustrative of the behavior on real datasets. So instead of having these toy examples, I'd suggest the authors have a thorough discussion on theoretical and experimental results.
> > >
> > > $ Response$: Thank you for your comments. We agree that the complex nature of real datasets is not fully reflected by our example. Our motivating example though could potentially help and inspire the readers who are unfamiliar with the literature. We have shortened and made concise the motivating example.

---

### Author Response · Authors · 2019-11-15
**We thank all the reviewers for their constructive comments.**

In the updated paper, the major changes are

1) The SENSE models have been tested again with stronger PGD attacks and the results are updated in Table 1 and Table3.
MNIST: PGD40 96.46%  -> PGD500 91.74%.
CIFAR: PGD20 65.17%  -> PGD100 57.23%.

The qualities of PGD attacks are checked by Figure 10 and Figure 11.

2) The reference of Suggala et al. (2018) is added in the related work, and the comparison is added in Section 3.

3) The sensitivity analysis on $c$ of CIFAR models with WideResNet and CNNs are added.

---

### Decision · Program_Chairs · 2019-12-19

**Decision:**

Reject

**Comment:**

Thanks for your detailed feedback to the reviewers, which clarified us a lot in many respects.
However, there is still room for improvement; for example, convergence to a good solution needs to be further investigated.
Given the  high competition at ICLR2020, this paper is unfortunately below the bar.
We hope that the reviewers' comments are useful for improving the paper for potential future publication.